# Single-molecule localization microscopy reveals STING clustering at the trans-Golgi network through palmitoylation-dependent accumulation of cholesterol

Haruka Kemmoku[1,9], Kanoko Takahashi[1,9], Kojiro Mukai[1,9], Toshiki Mori[2], Koichiro M. Hirosawa[3], Fumika Kiku[4], Yasunori Uchida[1], Yoshihiko Kuchitsu[1], Yu Nishioka[5], Masaaki Sawa[5], Takuma Kishimoto[6], Kazuma Tanaka[6], Yasunari Yokota[7], Hiroyuki Arai[4], Kenichi G. N. Suzuki[3,8] ✉ & Tomohiko Taguchi[1] ✉

Stimulator of interferon genes (STING) is critical for the type I interferon response to pathogen- or self-derived DNA in the cytosol. STING may function as a scaffold to activate TANK-binding kinase 1 (TBK1), but direct cellular evidence remains lacking. Here we show, using single-molecule imaging of STING with enhanced time resolutions down to 5 ms, that STING becomes clustered at the trans-Golgi network (about 20 STING molecules per cluster). The clustering requires STING palmitoylation and the Golgi lipid order defined by cholesterol. Single-molecule imaging of TBK1 reveals that STING clustering enhances the association with TBK1. We thus provide quantitative proof-of-principle for the signaling STING scaffold, reveal the mechanistic role of STING palmitoylation in the STING activation, and resolve the long-standing question of the requirement of STING translocation for triggering the innate immune signaling.

The innate immune system is an evolutionary ancient system to fight against invading pathogens[1,2]. It acts as the first line of defense, detecting the virus and bacteria through pattern recognition receptors (PRRs) and activating intracellular signaling cascades that promote the expression of proinflammatory cytokines, type I interferons, and other antiviral proteins that all coordinate the elimination of pathogens and infected cells. PRRs include Toll-like receptors, RIG-I-like receptors, nucleotide-binding domain and leucine-rich repeat-containing receptors, and C-type lectin receptors (CLRs)[3–5]. Furthermore, STING (also known as MITA, ERIS, MPYS or TMEM173)[6–9] has been shown to control

a sensing pathway that is essential for the innate immune response against double-stranded DNA (dsDNA) viruses. While the STING-mediated dsDNA-sensing signaling pathway is critical for successful protection against infections, dysregulated STING activity leads to the excessive production of inflammatory mediators with detrimental effects on surrounding cells and tissues[10]. Recent studies have revealed the critical role of STING in the pathogenesis of a number of autoinflammatory and neurodegenerative diseases, such as STING-associated vasculopathy with onset in infancy (SAVI)[11,12], COPA syndrome[13–18], Parkinson's disease[19], and amyotrophic lateral sclerosis[20,21].

[1]Laboratory of Organelle Pathophysiology, Department of Integrative Life Sciences, Graduate School of Life Sciences, Tohoku University, Sendai, Japan. [2]United Graduate School of Agricultural Science, Gifu University, Gifu, Japan. [3]Institute for Glyco-core Research (iGCORE), Gifu University, Gifu, Japan. [4]Department of Health Chemistry, Graduate School of Pharmaceutical Sciences, University of Tokyo, Tokyo, Japan. [5]Research and Development, Carna Biosciences, Inc., Kobe, Japan. [6]Division of Molecular Interaction, Institute for Genetic Medicine, Hokkaido University Graduate School of Life Science, Sapporo, Hokkaido, Japan. [7]Department of EECE, Faculty of Engineering, Gifu University, Gifu, Japan. [8]Division of Advanced Bioimaging, National Cancer Center Research Institute, Tokyo, Japan. [9]These authors contributed equally: Haruka Kemmoku, Kanoko Takahashi, Kojiro Mukai. ✉e-mail: suzuki.kenichi.b7@f.gifu-u.ac.jp; tomohiko.taguchi.b8@tohoku.ac.jp

STING is an ER-localized transmembrane protein. After binding to cyclic GMP-AMP (cGAMP)[22], which is generated by cGAMP synthase (cGAS)[23] in the presence of cytosolic dsDNA, STING translocates through the Golgi to the trans-Golgi network (TGN), where STING recruits TBK1 kinase from the cytosol[24,25] and triggers the type I interferon and proinflammatory responses through the activation of interferon regulatory factor 3 (IRF3) and nuclear factor-kappa B (NF-κB). The activation of TBK1 and IRF3 by STING requires the palmitoylation of STING on Cys88 and Cys91 and the Golgi lipid order[26], leading to the hypothesis that STING becomes clustered in the specific lipid microdomain at the TGN in palmitoylation-dependent manner and that the cluster facilitates the activation of TBK1 and IRF3 by recruiting multiple TBK1 and IRF3 molecules simultaneously onto it. In the present study, we examined whether STING formed the signaling-competent homocluster at the TGN by photoactivated localization microscopy (PALM) and direct stochastic optical reconstruction microscopy (dSTORM).

## Results

### Cluster formation of STING at the TGN

To quantitatively analyze the STING cluster, we performed live-cell PALM observation of mEos4b-STING. First, we performed data acquisition using purified mEos4b embedded in polyvinyl alcohol (PVA) on glass at 33 ms/frame and quantified the blinking and photoactivation based on the kinetic model[27] (Fig. 1a, b). This is the calibration for the actual experiment that is described later. The distribution of fluorescence intensity of mEos4b could be fitted with a single lognormal function based on both Akaike and Bayesian information criteria[28] (Fig. 1c), indicating that only the mEos4b monomer was observed. The measured probability distribution of the number of blinking times before photobleaching ($N_{blink}$) and the probability density function of $T_{on}$ were fitted with predicted geometric distribution (Fig. 1d) or single exponential function (Fig. 1e), respectively, which allowed us to estimate $<N_{blink}>$ ($=k_d/k_b$) to be 1.7.

The photo-switching behavior of mEos4b in PVA was similar to that in bacterial cells[29]. Nevertheless, we cannot exclude the possibility that PVA matrix restricts oxygen accessibility and exerts redox effects, and the imperative necessity arises to validate the methodology. Therefore, to test if we could accurately estimate the number of molecules in clusters using $<N_{blink}>$, we performed positive control observations. PC3 cells express caveolin but remain devoid of cavin1, thereby resulting in the absence of caveolae. A previous report[30] showed that the transfection of PC3 cells with cavin1 cDNA induced the formation of caveolae. Accordingly, we performed PALM observations of mEos4b-cavin1 in PC3 cells (Fig. 1f, left and middle), taking advantage of the 100% labeling efficiency of cavin1 with mEos4b. The images were binarized (Fig. 1f, right) using a traditional method of kernel density estimation (KDE)[31], which is applicable to a wide range of objects of observation[32]. By dividing the numbers of emission bursts of mEos4b in clusters with $1+<N_{blink}>$ for the unbiased multi-blinking correction[27], the number of cavin1 molecules in each caveolae (hereafter, denoted as [#cavin1/caveolae]) was determined. The average of [#cavin1/caveolae] was estimated to be $52 \pm 2$ (Fig. 1g, left), which is close to the previously reported one ($50 \pm 5$) measured by fluorescence correlation spectroscopy (FCS)[33]. Another segmentation method of the PALM images, SR-Tesseler[34], which has been frequently used[32], gave essentially the similar results (Fig. 1g, right). These results unequivocally affirm the successful quantification of molecular counts in clusters using $<N_{blink}>$ of mEos4b in PVA.

mEos4b-STING was stably expressed with virus transduction in STING-knockout (KO) mouse embryonic fibroblasts (MEFs). In these cells, the expression level of STING was comparable to that of endogenous STING in MEFs (Supplementary Fig. 1a). mEos4b-STING activated TBK1 and IRF3 with less potency than wild-type (WT) STING. We stimulated the cells with DMXAA (a membrane-permeable STING agonist) for the indicated time and obtained live-cell PALM images (Fig. 1h−k). The number of STING molecules in each cluster (hereafter, denoted as [#STING/cluster]) was determined and we found that the average of [#STING/cluster] increased up to 360 min after stimulation (Fig. 1i, j). The clusters consisting of >800 STING molecules formed around 30−60 min after stimulation (Fig. 1k). SR-Tesseler also gave essentially the similar results (Supplementary Fig. 2a). Importantly, STING clusters were also observed after stimulation with HT-DNA transfection (Supplementary Fig. 3).

The average of [#STING/cluster] estimated by performing the PALM data acquisitions at 5 ms/frame, was not significantly different from those at 33 ms/frame (Supplementary Fig. 4). We also corrected multi-blinking of mEos4b by the recently reported model-based correction (MBC) workflow, calibration-free estimation of blinking dynamics[35], and found no significant difference between two methods of multiple-blinking correction (Supplementary Fig. 5).

Phosphorylated TBK1 (p-TBK1) and phosphorylated STING (p-STING) emerged 30−60 min after stimulation in cells expressing mEos4b-STING (Supplementary Fig. 1b). Given that STING activates TBK1 at the TGN[26], not the other domains of the Golgi, these results suggested that mEos4b-STING reached the TGN by that time. To corroborate this, we performed simultaneous dual-color observation of PALM of mEos4b-STING and dSTORM of a cis-Golgi network (CGN) protein, GM130-HaloTag7, or a TGN protein, TGN38-HaloTag7 labeled with SaraFluor650B (SF650B) in living cells. As indicated by arrowheads (Fig. 2a), the mEos4b-STING clusters colocalized with GM130 10 min, not 30 min after stimulation. The quantitative coordinate-based colocalization (CBC) analysis[36] showed that the localization coordinates of mEos4b-STING and GM130-HaloTag7-SF650B were distributed in higher CBC scores (0.7−1.0) more frequently than control 10 min after stimulation, but were reduced to the control level 30 min after stimulation (Fig. 2b, c). In contrast, the mEos4b-STING clusters became extensively colocalized with TGN38 30−60 min after stimulation (Fig. 2d−f). The colocalization of mEos4b-STING with TGN38 then decreased during 60−120 min after stimulation. It is of note that Spearman's correlation coefficients pertaining to the amounts of mEos4b-STING or TGN38-HaloTag7 and the CBC scores 60 min after stimulation consistently fell within the range of −0.3−0.3 across all instances (Supplementary Fig. 6). These results indicated that the colocalization of mEos4b-STING with TGN38-HaloTag7 was less affected by the expression level of mEos4b-STING or TGN38-HaloTag7.

After the exit of STING from the Golgi/TGN, STING translocates to recycling endosomes[26,37]. In line with this, colocalizations of mEos4b-STING with recycling endosomal proteins (transferrin receptor (TfnR) and Rab11) increased only 120-360 min after DMXAA stimulation (Fig. 3a−f).

In sum, the PALM analysis (Fig. 1i, j) showed that STING became clustered at the TGN, with some clusters containing >800 STING molecules (Fig. 1k).

### Palmitoylation of STING is required for the cluster formation at the TGN

We previously demonstrated that STING palmitoylation, the post-translational modification essential for the STING signaling[26], facilitated the STING clustering to trigger the activation of TBK1 and IRF3. To prove this, we performed live-cell PALM observation of mEos4b-STING in the presence of palmitoylation inhibitors, 2-bromopalmitate (2-BP) and a STING-specific palmitoylation inhibitor H-151[38] (Fig. 4a, b). The treatment with both inhibitors decreased the average of [#STING/cluster] and suppressed the formation of clusters consisting of >800 STING molecules after stimulation with DMXAA or HT-DNA (Fig. 4c−f, Supplementary Figs. 2b, and 3). To further corroborate the role of STING palmitoylation in the STING clustering, we generated STING-KO MEFs stably expressing mEos4b-STING (C88/91S) (Fig. 4g), the

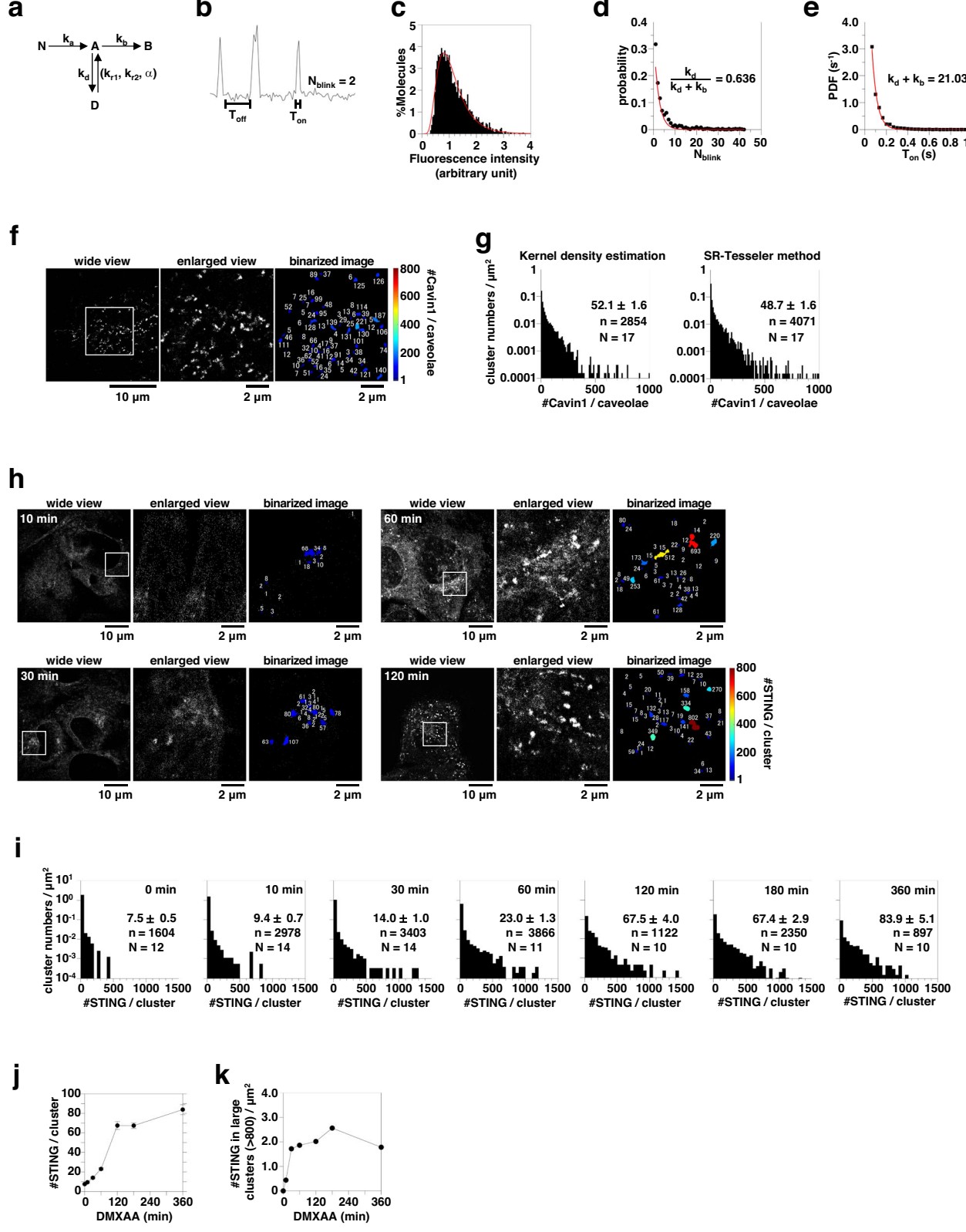

palmitoylation deficient variant of STING[26]. Live-cell PALM observation of mEos4b-STING (C88/91S) showed that the average of [#STING/cluster] of this variant was significantly decreased, compared to that of wild-type (WT) STING (Fig. 4h–j and Supplementary Fig. 2c). mEos4b-STING (C88/91S) was less active in forming the cluster containing >800 STING molecules (Fig. 4k).

## Cluster formation in the presence of the disease-causative COPA variants

COPA syndrome is a monogenic disorder of immune dysregulation characterized by the increased expression of type I interferon-stimulated genes. The disease is caused by heterozygous mutations of the COPA gene[13], encoding the α subunit (α-COP) of the COP-I

**Fig. 1 | Stimulation-dependent formation of STING cluster. a** Kinetic model of photoactivation and photobleaching of mEos4b. N, A, B, and D indicate nonactive, active, bleach, and dark states, respectively. $k_a$, $k_b$, $k_d$ indicate kinetic rates from N to A, from A to B, and from A to D, respectively. $k_{r1}$ and $k_{r2}$ indicate kinetic rates from D to A and α indicates the ratio between the contributions of the fast recovery rate $k_{r2}$ and the slow recovery rate $k_{r1}$. **b** A typical single-molecule emission trace allows us to estimate the kinetic rates. $T_{on}$ and $T_{off}$ indicate the fluorescence-on time before the molecule goes into the dark state or it photobleaches, and the fluorescence off-time, respectively. **c** Distribution of fluorescence intensity of individual mEos4b spots in PVA. The histogram could be fitted with a single log-normal function based on the Akaike and Bayesian information criteria (red curve). **d** Distribution of $N_{blink}$ fitted with a geometric distribution (red line). **e** $T_{on}$ distribution of mEos4b fitted with a single exponential decay (red line). **f** (left) Typical PALM super-resolution image of mEos4b-cavin1 in PC3 cells. (middle) Enlarged PALM image of the part indicated by the square in the left image. (right) Binarized PALM images after segmentation by kernel density estimation (KDE) method. The

numbers of mEos4b-cavin1 molecules/caveolae are indicated, and the clusters are colored according to the numbers. **g** The distribution of [#cavin1/caveolae]. KDE (left) or SR-Tesseler (right) method was used for segmentation. The number in the graph indicates mean ± SEM. n and N indicate the number of examined clusters and cells, respectively. **h** (left) Typical PALM super-resolution images of mEos4b-STING-expressing *Sting*[-/-] MEFs after DMXAA stimulation for indicated times. (middle) Enlarged PALM images of the part indicated by the square in the left images. (right) Binarized PALM images after segmentation by KDE method. The numbers of mEos4b-STING molecules/cluster are indicated, and clusters are colored according to the numbers. **i** The distribution of [#STING/cluster] in cells stimulated with DMXAA for the indicated times. The KDE method was used for the segmentation. The number in the graph indicates mean ± SEM. n and N indicate the number of examined clusters and cells, respectively. Time course of (**j**) the average of [#STING/cluster] (Data are presented as mean ± SEM) and (**k**) #STING in large clusters/μm² (large cluster = cluster consisting of >800 STING molecules) after DXMAA stimulation. Source numerical data are available in source data.

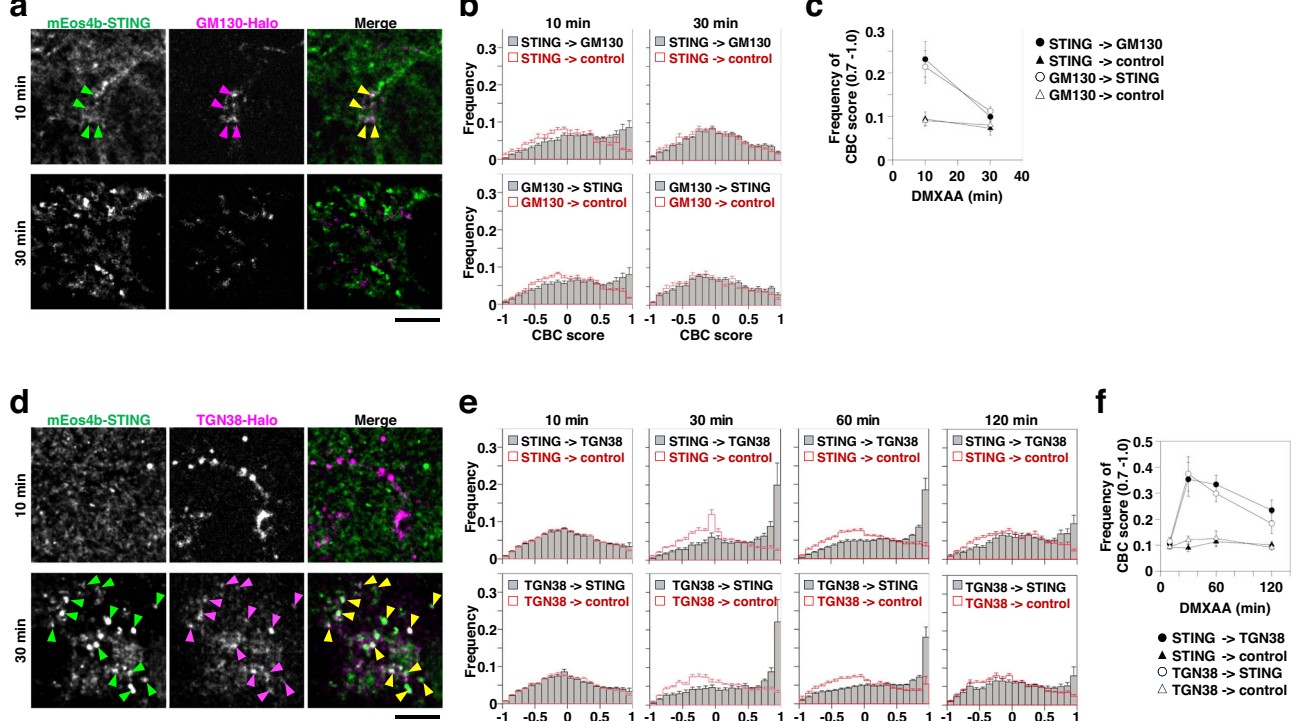

**Fig. 2 | The colocalization analysis of STING with the Golgi proteins. a** mEos4b-STING- and GM130-HaloTag7-expressing *Sting*[-/-] MEFs were stimulated with DMXAA for the indicated times. Typical PALM images of mEos4b-STING and dSTORM images of GM130-HaloTag7-SF650B. Colocalized areas are indicated by arrowheads. **b** Distribution of frequency of coordinate-based colocalization (CBC) scores for STING relative to GM130 (top) and for GM130 relative to STING (bottom). CBC scores relative to control (180° rotated image) are shown in red. The number of replicates was 10 (10 min) and 12 (30 min), respectively. Data are presented as mean ± SEM. **c** Time course of summation of frequency in each bin of CBC scores between 0.7 and 1.0 for STING relative to GM130 (closed circle), for GM130 relative to STING (open circle), for STING relative to control (closed triangle) and for GM130 relative to control (open triangle). Data are presented as mean ± SEM. **d** mEos4b-

STING- and TGN38-HaloTag7-expressing *Sting*[-/-] MEFs were stimulated with DMXAA for the indicated times. Typical PALM images of mEos4b-STING and dSTORM images of TGN38-HaloTag7-SF650B. Colocalized areas are indicated by arrowheads. **e** Distribution of frequency of CBC scores for STING relative to TGN38 (top) and for TGN38 relative to STING (bottom). CBC scores relative to control are shown in red. The number of replicates was 12 (10 min), 11 (30 min), 13 (60 min) and 12 (120 min), respectively. Data are presented as mean ± SEM. **f** Time course of summation of frequency in each bin of CBC scores between 0.7 and 1.0 for STING relative to TGN38 (closed circle), for TGN38 relative to STING (open circle), for STING relative to control (closed triangle) and for TGN38 relative to control (open triangle). Data are presented as mean ± SEM. Scale bars, 5 μm (**a** and **d**). Source numerical data are available in source data.

complex, which is essential for the retrograde transport from the Golgi to the ER and may regulate the anterograde transport from the ER to the Golgi. We and others have recently shown that STING was constitutively activated in the disease-causative COPA-expressing cells because of the inefficient retrograde transport of STING to the ER, and that the constitutive activation of STING

contributed to the upregulated inflammatory responses in COPA syndrome[14–18].

We stably expressed the disease-causative COPA variants in cells expressing mEos4b-STING and performed live-cell PALM observation. The average of [#STING/cluster] was significantly increased in the COPA variants-expressing cells, compared to control cells (Fig. 5 and

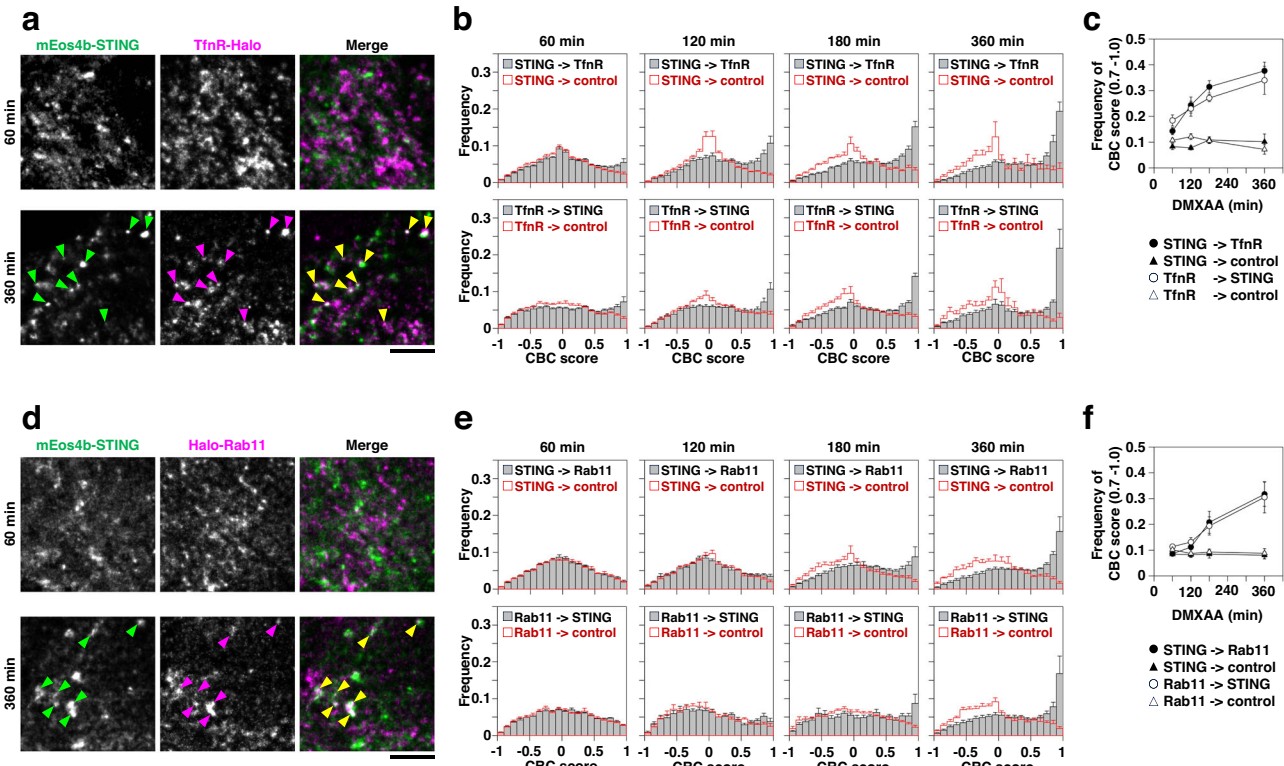

**Fig. 3 | The colocalization analysis of STING with recycling endosomal proteins.**
**a** mEos4b-STING- and TfnR-HaloTag7-expressing *Sting*[-/-] MEFs were stimulated with DMXAA for the indicated times. Typical PALM images of mEos4b-STING and dSTORM images of TfnR-HaloTag7-SF650B. Colocalized areas are indicated by arrowheads. **b** Distribution of frequency of coordinate-based colocalization (CBC) scores for STING relative to TfnR (top) and for TfnR relative to STING (bottom). CBC scores relative to control (180°rotated image) are shown in red. The number of replicates was 10 for all stimulation periods. Data are presented as mean ± SEM. **c** Time course of summation of frequency in each bin of CBC scores between 0.7 and 1.0 for STING relative to TfnR (closed circle), for TfnR relative to STING (open circle), for STING relative to control (closed triangle) and for TfnR relative to control (open triangle). Data are presented as mean ± SEM. **d** mEos4b-STING- and

HaloTag7-Rab11-expressing *Sting*[-/-] MEFs were stimulated with DMXAA for the indicated times. Typical PALM images of mEos4b-STING and dSTORM images of HaloTag7-SF650B-Rab11. Colocalized areas are indicated by arrowheads. **e** Distribution of frequency of CBC scores for STING relative to Rab11 (top) and for Rab11 relative to STING (bottom). CBC scores relative to the control are shown in red. The number of replicates was 10 for all stimulation periods. Data are presented as mean ± SEM. **f** Time course of summation of frequency in each bin of CBC scores between 0.7 and 1.0 for STING relative to Rab11 (closed circle), for Rab11 relative to STING (open circle), for STING relative to control (closed triangle) and for Rab11 relative to control (open triangle). Data are presented as mean ± SEM. Scale bars, 5 μm (**a** and **d**). Source numerical data are available in source data.

Supplementary Fig. 2d), showing that STING formed the clusters in the COPA variant-expressing cells. Next we examined whether cGAMP, the natural STING ligand, was required for the clustering of STING. To address this question, we exploited cGAS-KO MEFs[16], in which the production of cGAMP was inhibited. As shown (Supplementary Fig. 7), the average of [#STING/cluster] in cGAS-KO MEF cells expressing COPA(D243G) was significantly larger than that in control cells. These results showed that cGAMP binding to STING was not intrinsically essential for the STING clustering.

**The role of cholesterol in the STING signaling and clustering**
We previously showed that the Golgi lipid order, besides STING palmitoylation, was essential for the STING signaling[26]. Treatment of cells with D-ceramide-C6, which caused a reduced membrane order in the Golgi membrane through the generation of short-chain sphingomyelin[39], suppressed the STING signaling (Fig. 6a) and decreased the average of [#STING/cluster] (Fig. 6b–d and Supplementary Fig. 2e). In contrast, L-ceramide-C6, a non-metabolizable enantiomer of D-ceramide-C6, had a minimal effect on the average of [#STING/cluster].

Di-4-ANEPPDHQ is a fluorescent probe that exhibits distinct changes in its emission spectrum in response to variations in lipid environment[40]. Notably, it is highly sensitive to the state of lipid packing within cellular membranes[41]. An increase in the concentration

of cholesterol in biomembranes enhances lipid packing, thus affecting the emission spectrum of Di-4-ANEPPDHQ. With this probe, we confirmed that D-ceramide-C6, not L-ceramide-C6, caused a reduced membrane order in the Golgi membrane (Supplementary Fig. 8).

We sought to examine a role of cholesterol, a lipid essential for the generation of the lipid order[42,43], in the STING signaling and clustering by interfering with the function of oxysterol-binding protein (OSBP). OSBP transports cholesterol from the ER to the TGN[44], and its function is inhibited by 25-hydroxycholesterol (25-HC)[45] or a plant-derived steroidal saponin OSW-1[46]. Treatment of cells with these compounds (30 μM of 25-HC for 16 h or 0.125 nM of OSW-1 for 6 h) suppressed the stimulation-dependent emergence of p-TBK1, p-STING, and phosphorylated IRF3 (p-IRF3) (Fig. 7a, b), without affecting the translocation of STING from the ER to the Golgi (Supplementary Fig. 9a–c). Treatment of cells with 25-HC or OSW-1 decreased the average of [#STING/cluster] (Fig. 7c–e and Supplementary Fig. 2f) and suppressed the formation of clusters consisting of >800 STING molecules (Fig. 7f).

We have recently developed a cell-free assay[47] using recombinant TBK1 and microsomes harboring STING, so that we can examine whether STING is primed for the phosphorylation by TBK1. When microsomes were prepared from cells stimulated with DMXAA and used for the reaction, STING was phosphorylated[47] (Fig. 7g, h). In contrast, STING was insufficiently phosphorylated when microsomes

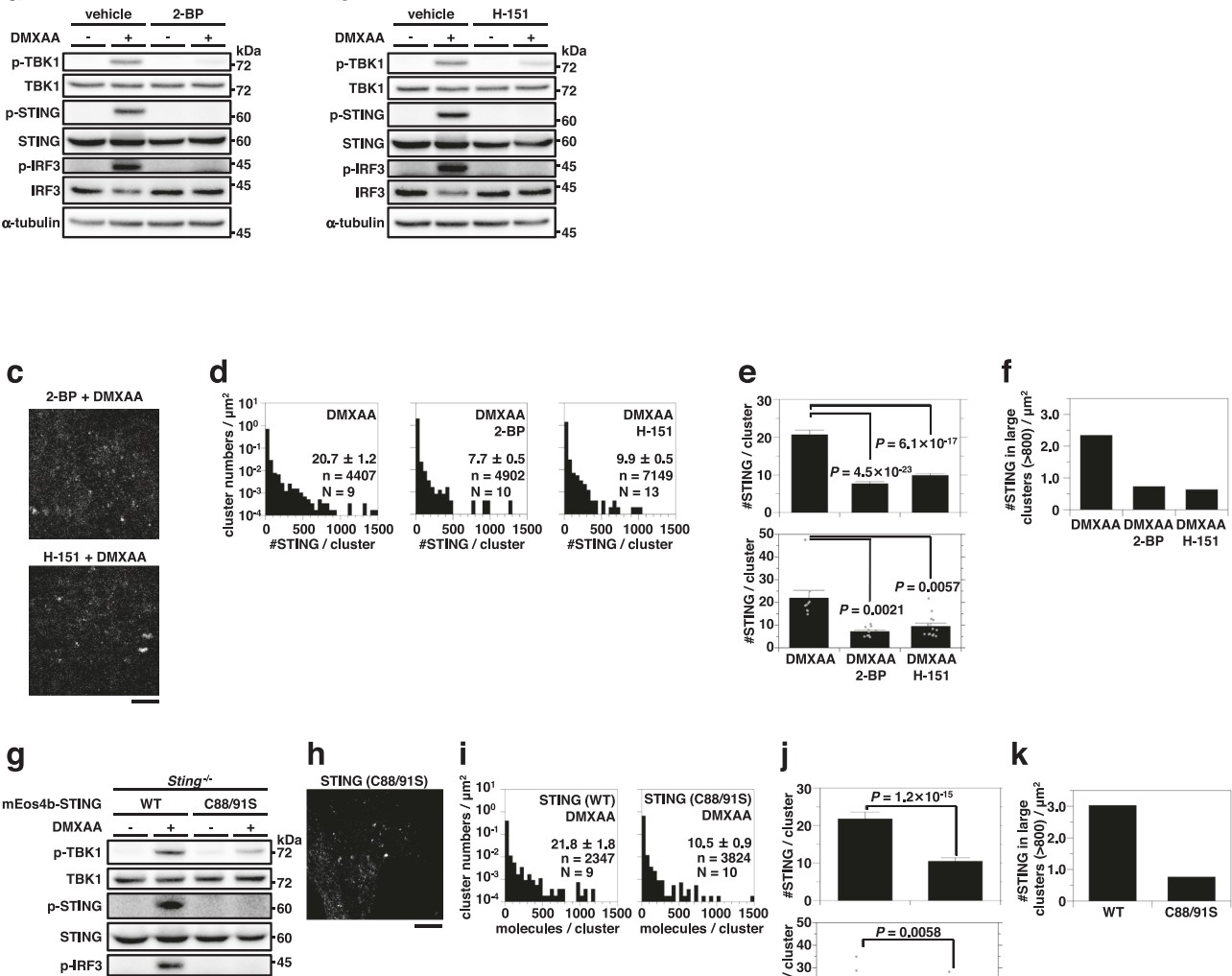

**Fig. 4 | Palmitoylation of STING is required for clustering.** mEos4b-STING-expressing *Sting*[-/-] MEFs were pretreated with (**a**) 2-bromopalmitate (2-BP) (200 μM) for 4 h or (**b**) H-151 (10 μM) for 2 h, and stimulated with DMXAA. Cell lysates were prepared and analyzed by western blot. **c** Typical PALM images of mEos4b-STING in cells treated with (top) 2-BP or (bottom) H-151 after DMXAA stimulation for 60 min. **d** The distribution of [#STING/cluster] in (**c**). The KDE method was used for the segmentation. n and N indicate the number of examined clusters and cells, respectively. **e** The average of [#STING/cluster] (top: average in all the observed clusters, bottom: average of the averages in individual cells shown by individual dots, data are presented as mean ± SEM) and (**f**) #STING in large clusters/μm² (large cluster = cluster consisting of >800 STING molecules) in (**d**). The *P*-values (Welch's *t*-test, both-sided) were less than the significance levels

corrected by the Holm-Sidak method. **g** mEos4b-STING (WT or C88/91 S)-expressing *Sting*[-/-] MEFs were stimulated with DMXAA for 60 min. Cell lysates were prepared and analyzed by western blot. **h** Typical PALM images of mEos4b-STING (C88/91 S) in cells stimulated with DMXAA for 60 min. **i** The distribution of [#STING/cluster] in (**h**). The KDE method was used for the segmentation. n and N indicate the number of examined clusters and cells, respectively. **j** The average of [#STING/cluster] (top: average in all the observed clusters, bottom: average of the averages in individual cells shown by individual dots, data are presented as mean ± SEM) and (**k**) #STING in large clusters/μm² (large cluster = cluster consisting of >800 STING molecules) in (**i**). The *P*-values (Welch's *t*-test, both-sided) were less than the significance level. Scale bars, 5 μm (**c** and **h**). Source numerical data and unprocessed blots are available in source data.

from cells pretreated with 25-HC or OSW-1 were used. Importantly, the treatment of the microsomes with cholesterol/methyl-β-cyclodextrin (MβCD) complex, not MβCD alone, restored the phosphorylation of STING.

Lastly, we performed cholesterol add-back experiments in live cells. STING-KO MEFs stably expressing mEos4b-STING were pretreated with 25-HC or OSW-1, incubated with medium containing cholesterol-methyl-β-cyclodextrin complex for 3 h, and then stimulated with DMXAA for 60 min. As shown (Supplementary Fig. 9d, e), the cholesterol add-back to cells restored the formation of p-TBK1, p-STING, and p-IRF3.

In sum, these results (Fig. 7 and Supplementary Fig. 9) suggested the role of cholesterol in the STING signaling and clustering.

## The STING cluster colocalizes with cholesterol-enriched domains

The role of cholesterol in the STING signaling and clustering led us to examine the subcellular localization of cholesterol. To visualize cholesterol, we exploited a recently established cholesterol biosensor iD4H (derived from the *Clostridium Perfringens* theta-toxin)[48]. Recombinant mNeonGreen-tagged iD4H bound to liposomes at which cholesterol exceeded 20% molar content (Supplementary Fig. 10a, b).

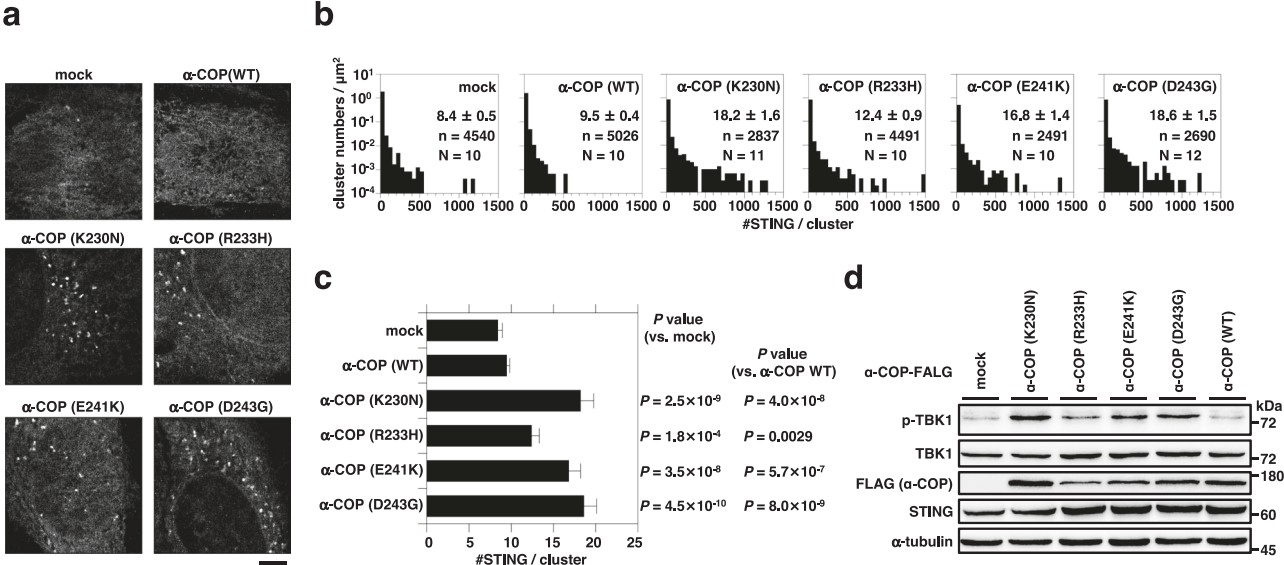

**Fig. 5 | Expression of the α-COP variants induces the clustering of STING.**
**a** Typical PALM images of *Sting^-/-* MEFs expressing mEos4b-STING and/or α-COP-FLAG (WT or the disease-causative variants). **b** The distribution of [#STING/cluster] in (**a**). The KDE method was used for the segmentation. n and N indicate the number of examined clusters and cells, respectively. **c** The average of [#STING/cluster] in (**a**). Data are presented as mean ± SEM. The *P*-values (Welch's *t* test, both-sided) were less than the significance levels corrected by the Holm-Sidak method. **d** Lysates of cells in (**a**) were prepared and analyzed by western blot. Scale bar, 5 μm (**a**). Source numerical data and unprocessed blots are available in source data.

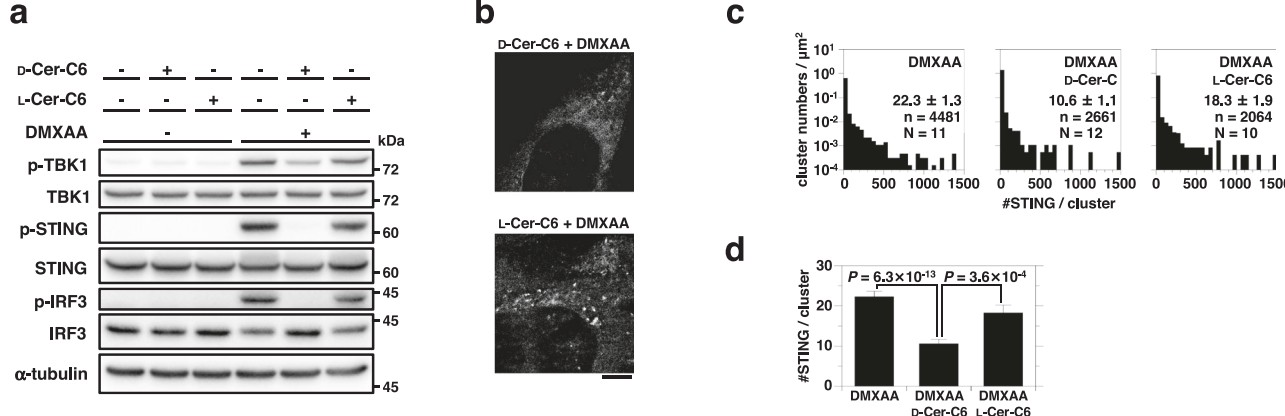

**Fig. 6 | Disturbing the Golgi lipid order with D-ceramide-C6 suppresses STING clustering. a** *Sting^-/-* MEFs expressing mEos4b-STING were treated with D-ceramide-C6 (D-Cer-C6) (20 μM) or L-ceramide-C6 (L-Cer-C6) (20 μM) for 60 min. Cells were then stimulated with DMXAA for 60 min. Cell lysates were prepared and analyzed by western blot. **b** Typical PALM images of mEos4b-STING in cells treated with D-ceramide-C6 or L-ceramide-C6, followed by DMXAA stimulation for 60 min. **c** The distribution of [#STING/cluster] in (**b**). The KDE method was used for the segmentation. n and N indicate the number of examined clusters and cells, respectively. **d** The average of [#STING/cluster] in (**c**). Data are presented as mean ± SEM. The *P*-values (Welch's t-test, both-sided) were less than the significance levels corrected by the Holm-Sidak method. Scale bar, 5 μm (**b**). Source numerical data and unprocessed blots are available in source data.

In the resting state, mRuby3-STING localized at the ER and did not colocalize with puncta positive with mNeonGreen-iD4H (Fig. 7i, j). In contrast, after STING stimulation with DMXAA, mRuby3-STING, especially the one forming puncta at the Rab6 (a TGN protein)-positive area, partly colocalized with mNeonGreen-iD4H. We also noted the stimulation-dependent increase of the signal of mNeonGreen-iD4H at the Rab6-positive area (Fig. 7k).

The increased signal of mNeonGreen-iD4H did not colocalize with lipid droplets visualized by Lipi-Blue (Supplementary Fig. 11), suggesting that lipid droplet was not involved in the enhanced accumulation of mNeonGreen-iD4H at the TGN.

The treatment with OSW-1 suppressed the increase of mNeonGreen-iD4H at the Rab6-positive area (Supplementary Fig. 10c, d). The area of the Rab6-positive region was not affected with OSW-1

treatment (Supplementary Fig. 10e). Intriguingly, palmitoylation inhibitors (2-BP and H-151) abolished the increase of the signal of mNeonGreen-iD4H at the Rab6-positive area (Fig. 7i–k). These results indicated the STING palmitoylation drove the local accumulation of cholesterol in the cytosolic leaflet of the TGN membranes.

We also examined the subcellular distribution of cholesterol by filipin, a polyene macrolide antibiotic. As shown (Supplementary Fig. 12), DMXAA stimulation did not increase the signal of filipin at the Rab6-positive area. Thus, the level of the cholesterol in the TGN membranes was not grossly affected by STING stimulation.

After STING stimulation, mNeonGreen-iD4H colocalized also well with TBK1-mScarletI (Supplementary Fig. 10f). The treatment with OSW-1 partially suppressed the TBK1-positive puncta formation, which was also indicated by the remaining cytoplasmic signal of TBK1. The

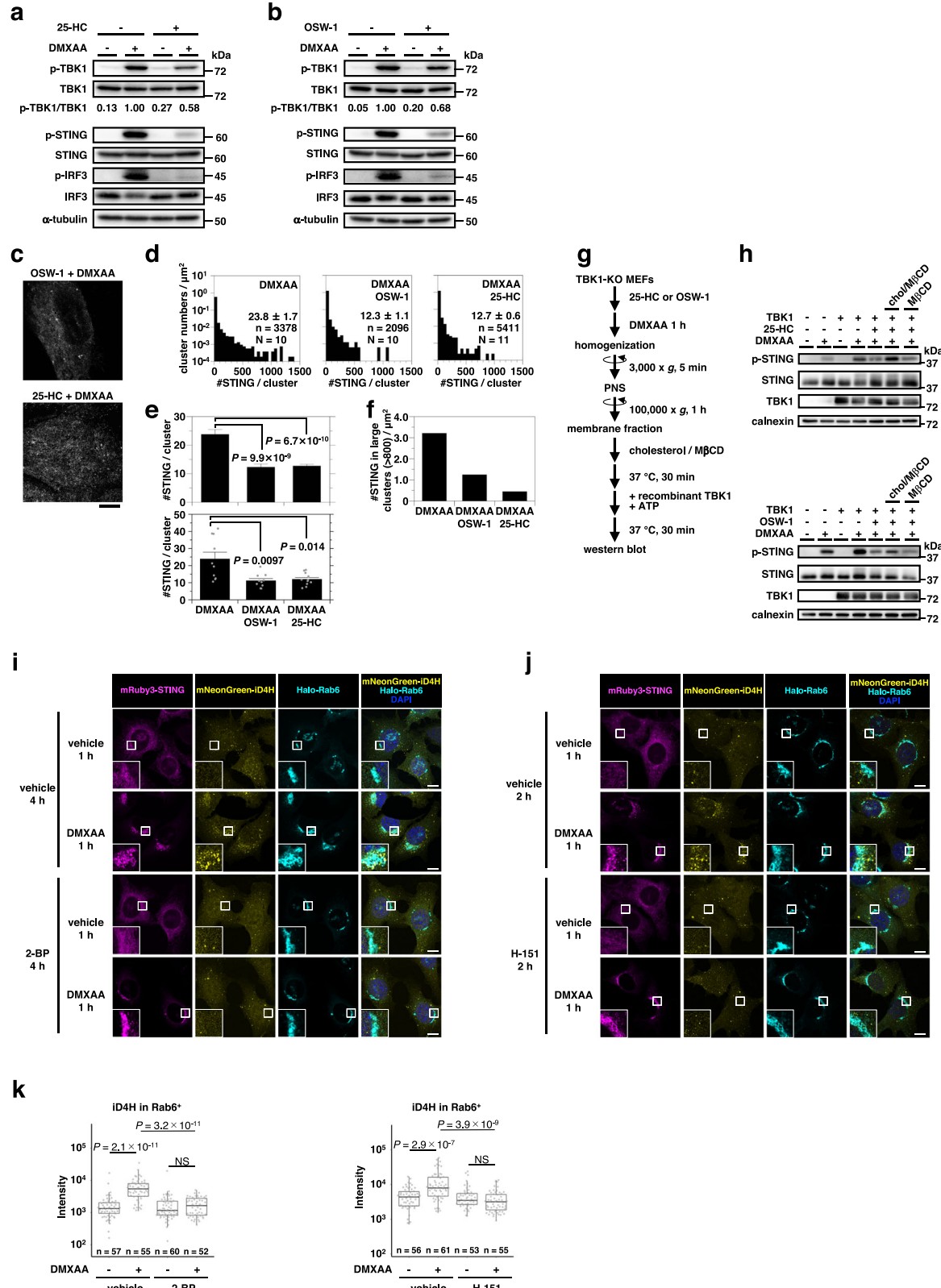

colocalization of the TBK1-positive puncta with iD4H motivated us to examine the lipid order of the Golgi membranes. We observed that the TBK1-positive puncta showed a high value of the generalized polarization (GP) of di-4-ANEPPDHQ (movie S1). These results suggested that the membrane domains positive with TBK1 had high membrane rigidity or were highly ordered.

## The clustering of STING facilitates TBK1 recruitment

STING has long been assumed to act as a protein scaffold to recruit and activate TBK1[49]. To directly examine if the clustering of STING facilitates TBK1 recruitment from the cytosol, simultaneous dual-color live-cell PALM and single-molecule imaging were performed. We observed that single molecules of TBK1-HaloTag7-SF650T molecules were

**Fig. 7 | Cholesterol in the TGN facilitates the clustering of STING.** *Sting*[-/-] MEFs expressing mEos4b-STING were pretreated with (**a**) 25-HC (30 μM) for 16 h or (**b**) OSW-1 (0.125 nM) for 6 h, and stimulated with DMXAA for 60 min. Cell lysates were prepared and analyzed by western blot. **c** Typical PALM images of mEos4b-STING in cells pretreated with (top) OSW-1 (0.125 nM) for 6 h or (bottom) 25-HC (30 μM) for 16 h, followed by stimulation with DMXAA for 60 min. **d** The distribution of [#STING/cluster] in (**c**). The KDE method was used for the segmentation. n and N indicate the number of examined clusters and cells, respectively. **e** The average of [#STING/cluster] (top: average in all the observed clusters, bottom: average of the averages in individual cells shown by individual dots, data are presented as mean ± SEM) and (**f**) #STING in large clusters/μm² (large cluster = cluster consisting of >800 STING molecules) in (**d**). The *P*-values (Welch's *t*-test, both-sided) were less than the significance levels corrected by the Holm-Sidak method. **g** Scheme of TBK1-dependent phosphorylation of STING in vitro. **h** TBK1-knockout MEFs were pretreated with (top) 25-HC (30 μM) for 16 h or (bottom) OSW-1 (0.125 nM) for 6 h, followed by stimulation with DMXAA for 60 min. The membrane fraction of the cells was collected by ultracentrifugation. The resuspended membrane fraction was incubated with cholesterol-methyl-β-cyclodextrin complex (chol/MβCD) or only methyl-β-cyclodextrin (MβCD), followed by incubation with recombinant TBK1 and ATP. Phosphorylation of STING at Ser365 was then analyzed by western blot. **i**, **j** *Sting*[-/-] MEFs stably expressing mRuby3-STING (magenta), mNeonGreen-iD4H (yellow), and HaloTag7-Rab6 (cyan) were pretreated with (**i**) 2-BP (200 μM) for 4 h or (**j**) H-151 (10 μM) for 2 h, and then stimulated with DMXAA for 60 min. Cells were fixed, permeabilized, and stained with HaloTag SaraFluor 650 T Ligand and with DAPI (blue). **k** The fluorescence intensity of mNeonGreen-iD4H within the Rab6-positive region in cells observed in (**i** and **j**) was quantified. Data are presented in box-and-whisker plots with the minimum, maximum, sample median, first versus third quartiles and whiskers extend to a maximum of 1.5× interquartile range beyond the box. The data were statistically analyzed by performing one-way analysis of variance followed by Tukey-Kramer post hoc test for multiple comparisons. The sample size (n) represents the number of cells examined over 3 independent experiments. NS, not significant. Scale bars, 5 μm (**c**) and 10 μm (**i** and **j**). Source numerical data and unprocessed blots are available in source data.

transiently recruited to mEos4b-STING clusters after STING stimulation (Fig. 8a−c, Supplementary Fig. 1c, and movie S2). The lifetime of the colocalization durations of single-molecules of TBK1-HaloTag7-SF650T with STING clusters was determined as around 150 ms in DMXAA-stimulated cells (Fig. 8d, e). The result indicated that the large TBK1 clusters shown in Supplementary Fig. 10f were formed by transient recruitment of TBK1 to STING clusters. Treatment of cells with H-151, which reduced the size of STING cluster (Fig. 4c−f), significantly reduced it to 74 ms (Fig. 8d, e). Treatment of cells with MRT67307 (TBK1 inhibitor)[50] reduced it to 92 ms (Fig. 8d, e), but did not affect the size of STING cluster (Supplementary Fig. 13a, b). STING (L373A) lacks the ability to bind TBK1[51,52]. STING (L373A) clustered as wild-type STING (Supplementary Fig. 13b, c), however, the lifetime of colocalization durations of TBK1 with clusters of STING (L373A) was very short and comparable to that of control (Fig. 8d, e).

We analyzed the numbers of STING molecules within 100 nm radius around TBK1-HaloTag7-SF650T spots and those around randomized TBK1 spots in silico. The histograms of the difference showed that the frequency of TBK1 recruitment (Fig. 8f) and the density of STING around the TBK1 spots (Fig. 8g) were significantly reduced after treatment of cells with H-151 or MRT67307, or in cells expressing STING (L373A).

These results suggested that the recruitment of TBK1 to STING was facilitated by a palmitoylation-dependent STING cluster growth, and that the phosphorylation of STING by TBK1, in turn, stabilizes the binding of TBK1 to STING clusters.

## Discussion

By in vitro and in silico analyses, STING is suggested to oligomerize in the presence of cGAMP[51–53], which may facilitate the activation of the STING signaling. However, these results do not explain the reason why TBK1/IRF3 activation occurs at the Golgi/TGN[26,54–56], not at the ER where STING binds cGAMP. On the basis of the present cellular data by single-molecule localization microscopy, we propose a model underlying STING activation at the TGN (Fig. 8h and Supplementary Fig. 13d): Binding of cGAMP to STING dimer induces conformational change and/or oligomerization of STING[57], which allows STING to translocate from the ER to cis-Golgi network. STING further translocates to the TGN along the exocytic membrane flow, during which STING undergoes palmitoylation[26]. At the TGN, STING clusters with the aid of the palmitoyl groups on Cys88 and Cys91 on STING (Fig. 4), cholesterol (a lipid directly transported from the ER to the TGN by lipid-transfer proteins[58] (Fig. 7), and sphingomyelin (a lipid synthesized at the TGN)[39] (Fig. 6). Palmitoylation of STING drove the local accumulation of cholesterol in the cytosolic leaflet of the TGN membranes (Fig. 7), which may further facilitate STING clustering given the affinity of palmitoylated proteins and raft lipids[59].

By the analysis of the durations and frequencies of colocalization of TBK1 with STING (Fig. 8), we suggested that STING clustering facilitated TBK1 recruitment. The advantage of STING clustering in TBK1 recruitment may be explained as follows: Once TBK1 is detached from STING, there are other TBK1-free STING molecules around, in addition to the STING molecule that TBK1 originally binds to. Thus, STING clustering increases the encounter probability of TBK1 and STING. Clustering of STING may also be beneficial for TBK1 activation because two TBK1 dimers need to be placed close each other for autophosphorylation/autoactivation.

We noted that p-TBK1 was less affected in the conditions where the Golgi cholesterol was reduced (Fig. 7a, b). In contrast, p-TBK1 was severely reduced in the conditions where palmitoylation of STING was inhibited (Fig. 4a, b). We assume that orientation of individual STING molecules in the small STING cluster may differ in the two conditions. In cholesterol-depleted conditions, even a small STING cluster may accommodate two TBK1 dimers simultaneously, thus still facilitating autophosphorylation of TBK1. In contrast, a small cluster that palmitoylation-deficient STING forms may not accommodate two TBK1 dimers.

The TGN functions as the central hub for secretory cargo protein sorting and trafficking to the plasma membrane[60]. A number of studies have established a cholesterol- and/or sphingolipid-based system for cargo sorting and membrane-carrier biogenesis at the TGN[61]. In the present study, we showed that a cholesterol-based system at the TGN operated also in triggering the innate immunity signaling. By exploiting a cholesterol biosensor and an environmentally sensitive probe for lipid membranes, we further demonstrated that cholesterol generated a specific membrane domain, or a liquid-ordered domain, in which palmitoylated STING formed clusters to activate TBK1 at the TGN. Our results provided direct evidence of the generation of liquid-ordered lipid microdomain in the TGN, which was originally postulated as "lipid raft" by Simons' group 26 years ago[62]. Our results also endorsed in vitro observations that palmitoylation of proteins increased the affinity to liquid-ordered phase[39].

STING activates a wide array of innate immune and proinflammatory genes in addition to the type I interferons. However, the significance of the activation of these genes remains mostly elusive. One example is cholesterol 25-hydroxylase (CH25H)[63], which metabolizes cholesterol to 25-HC[64]. Given the inhibitory effect of 25-HC on the STING signaling, the induction of CH25H in response to STING activation may contribute to the homeostasis of the STING signaling. Of note, knockdown of CH25H in B16 melanoma cells enhanced the STING signaling (Supplementary Fig. 14), suggesting the causal link between the STING signaling and CH25H/25-HC.

H-151, a small-molecule inhibitor that suppresses STING palmitoylation on Cys91, has been developed[38]. Given the success of

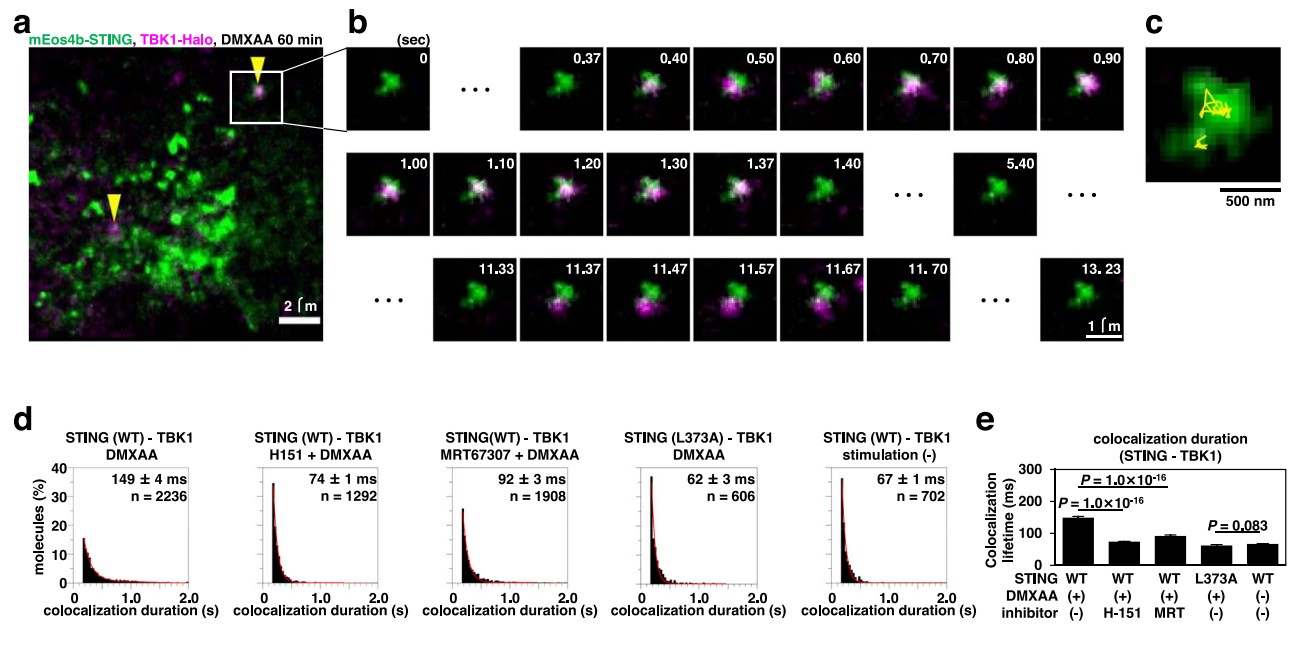

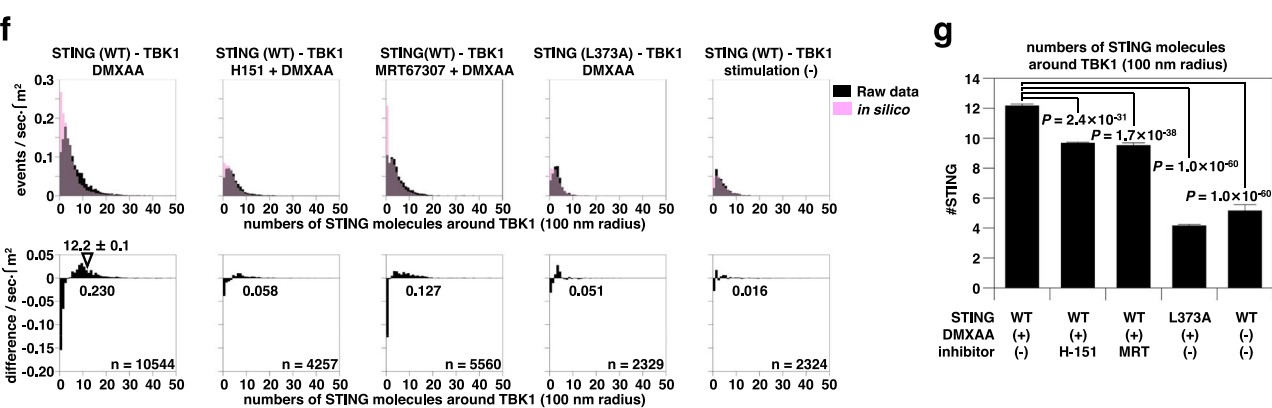

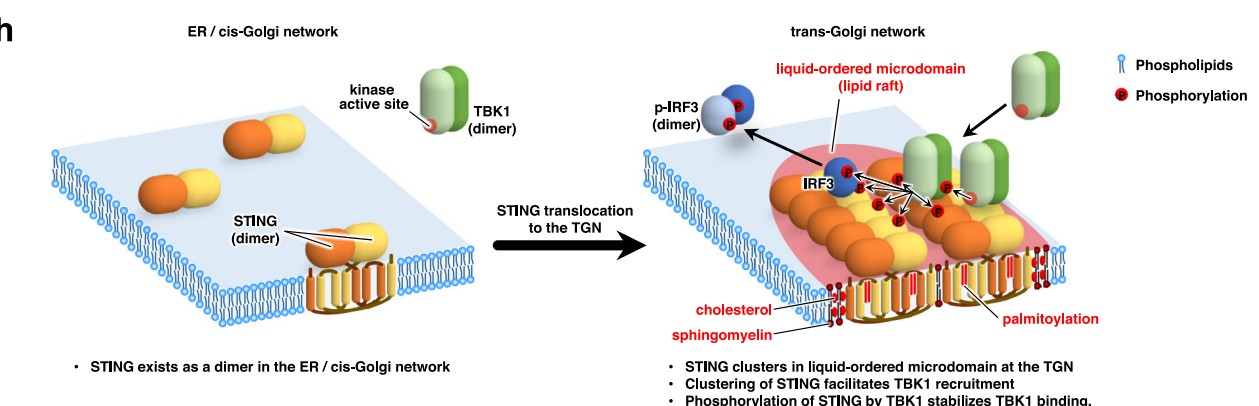

H-151 in quelling a variety of the STING-mediated inflammatory responses in animal models and the critical roles of the STING clustering in the STING signaling[65], the interference of the cellular cholesterol levels in particular, and ideally the ones in the TGN, may also provide a strategy to treat the STING-mediated inflammatory diseases.

## Methods

### Antibodies

Antibodies used in this study were as follows: rabbit anti-STING (19851-1-AP, dilution 1:1000) and rabbit anti-calnexin (10427-2-AP, dilution 1:1000) (Proteintech); rabbit anti-phospho-STING (D8F4W, dilution 1:1000 for western blot), rabbit anti-phospho-TBK1 (D52C2, dilution

**Fig. 8 | The recruitment of TBK1 requires the clustering of STING.** mEos4b-STING and TBK1-HaloTag7 were stably expressed in STING/TBK1-double knockout MEFs. **a** A typical overlapped image obtained by simultaneous observation of mEos4b-STING clusters (green, PALM image) and single-molecules of TBK1-HaloTag7-SF650T (magenta) 60 min after stimulation with DMXAA. Arrowheads indicate the colocalization of STING clusters with single molecules of TBK1. **b** The area indicated by square in (**a**) was expanded, and the image sequence is shown. The numbers at the top-right indicate time (second) from the first image. **c** Superposition of PALM image of a STING cluster and trajectories of TBK1-HaloTag7-SF650T. **d** Distribution of periods of colocalization between STING (WT and L373A mutant) clusters and single molecules of TBK1 in cells treated with H-151 (10 μM, 2 h) or MRT67307 (10 μM, 2 h) before stimulation with DMXAA for 60 min. The PALM image of STING clusters after DMXAA stimulation was superimposed with the image of single molecules of TBK1 in the steady-state cells, and the colocalization lifetime was estimated. n indicates the number of colocalizations between STING clusters and single molecules of TBK1HaloTag7-SF650T. **e** The colocalization lifetimes in (**d**). Data are presented as mean ± SEM. The P-values (log-rank test) in the comparison of colocalization lifetimes before and after treatment with H-151 or MRT67307, were less than the significance levels corrected by the Holm-Sidak method, whereas that of L373A mutant was not significantly different from the control. **f** (top) Distribution of the numbers of STING molecules within 100 nm radius around TBK1 spots (black bars) and those around randomized TBK1 spots in silico (magenta bars). Event numbers were normalized by the observation periods and areas. (bottom) The histograms show the difference between black bars and magenta bars in the top row. The number in the graphs indicates the summation of the differences in the range showing the continuous positive values. **g** The average number of STING molecules in the range showing continuous positive values of the difference in the bottom row in (**f**). Data are presented as mean ± SEM. The P-values (Welch's t-test, both-sided) were less than the significance levels corrected by the Holm-Sidak method. **h** A graphical abstract illustrating cholesterol- and palmitoylation-dependent STING clustering at the TGN. Source numerical data are available in source data.

1:1000), rabbit anti-IRF3 (D83B9, dilution 1:1000), and rabbit anti-phospho-IRF3 (4D4G, dilution 1:1000) (Cell Signaling Technology); rabbit anti-TBK1 (ab40676, dilution 1:1000) (Abcam); mouse anti-α-tubulin (10G10, dilution 1:1000) and mouse anti-FLAG (1E6, dilution 1:1000) (Wako); mouse anti-CH25H (J2617, dilution 1:100; Santa Cruz); Goat anti-Rabbit IgG (H + L) Mouse/Human ads-HRP (4050-05, dilution 1:10,000) and Goat anti-Mouse IgG (H + L) Human ads-HRP (1031-05, dilution 1:10,000) (SouthernBiotech); sheep anti-TGN38 (AHP499G, dilution 1:200) (Bio-Rad); mouse anti-GM130 (610823, dilution 1:4000) (BD Biosciences); Alexa 568-, 594-, or 647-conjugated secondary antibodies (A10037, A11016, A21448, dilution 1:1000) (Thermo Fisher Scientific).

## Reagents

The following reagents were purchased from the manufacturers as noted: DMXAA (14617, Cayman); di-4-ANEPPDHQ (D36802, Thermo Fisher Scientific); HaloTag® SaraFluor™ 650 T Ligand (A308-02, Goryo Chemical, Inc.), HaloTag® SaraFluor™ 650B Ligand (A201-01, Goryo Chemical, Inc.), D-ceramide-C6 (62525, Cayman); L-ceramide-C6 (24388, Cayman); MRT67307 (19916, Cayman); 2-Bromopalmitate (320-76562, Wako); HT-DNA (D6898, Sigma); Filipin III (70440, Cayman); 25-hydroxycholesterol (11097, Cayman); OSW-1 (30310, Cayman); Cholesterol, Water Soluble (C4951, Sigma); Dioleoyl phosphatidylcholine (850375, Avanti Polar Lipids); Cholesterol (C8667, Sigma); Lipi-Blue (LD01, DOJINDO). DMXAA was routinely used at 25 μg mL$^{-1}$ throughout the present study.

## PCR cloning

N-terminal mNeonGreen-, mRuby3-, or mEos4b-tagged mouse STING (NM_028261) was introduced into pMXs-IPuro. C-terminal HaloTag7-tagged mouse TBK1 (NM_019786), mouse GM130 (NM_001080968.2), mouse TfnR (Transferrin Receptor) (NM_011638.4), or mouse TGN38 (NM_009443.3) was introduced into pMXs-IBla. C-terminal mScarletI-tagged human TBK1 (NM_013254) was introduced into pMXs-IBla. N-terminal HaloTag7-tagged mouse Rab6a (NM_024287) or mouse Rab11a (NM_017382.5) was introduced into pMXs-IHyg. C-terminal FLAG-tagged mouse α-COP (NM_009938.4) was introduced into pMXs-IBla. The cDNA plasmid encoding N-terminal mEos4b-tagged mouse cavin1 (NM_008986.2) was produced by replacing GFP in the pEGFP-C1 vector (addgene, #68401) with mEos4b.

The iD4H (D4$^{Y415A/D434S/A463W}$) mutant fragment was generated by the standard two-step PCR mutagenesis technique using pCold I-GFPenvy-D4H (D4$^{D434S}$)[66] as a template. The iD4H fragment was introduced into pMXs-IBla. To construct E. coli expression vector, mNeonGreen-iD4H fragment was amplified by PCR using pmNeonGreen-iD4H as a template and cloned into pCold I (Takara Bio, Shiga, Japan).

## Cell culture

MEFs were cultured in DMEM supplemented with 10% fetal bovine serum (FBS) and penicillin/streptomycin/glutamine (PSG) in a 5% CO$_2$ incubator. MEFs that stably express tagged proteins were established using retrovirus. Plat-E cells were transfected with pMXs vectors, and the medium that contains the retrovirus was collected. MEFs were incubated with the medium and then selected with puromycin (2 μg mL$^{-1}$), blasticidin (5 μg mL$^{-1}$), or hygromycin (400 μg mL$^{-1}$) for several days. STING/TBK1- or STING/cGAS-double knockout MEFs were generated by CRISPR-Cas9 system[16,25]. Human prostate cancer (PC3; ATCC, CRL-1435) cells were cultured in HAM's F12 supplemented with 10% FBS, 100 μ/ml penicillin, and 100 μg/ml streptomycin in a 5% CO2 incubator.

## Immunocytochemistry

Cells were fixed with 4% paraformaldehyde (PFA) in PBS at room temperature for 15 min and permeabilized with 0.1% Triton X-100 in PBS at room temperature for 5 min. After blocking with 3% BSA in PBS, cells were incubated with primary antibodies. After washing with PBS three times, cells were then incubated with the secondary antibody at room temperature for 60 min, washed, and mounted with ProLong™ Glass Antifade Mountant (P36982, Thermo Fisher Scientific).

For cholesterol staining, cells were fixed with 4% paraformaldehyde in PBS at room temperature for 15 min. After permeabilization by freeze-thawing with liquid nitrogen, cells were incubated with filipin III (50 μg mL$^{-1}$) at room temperature for 30 min. For HaloTag7 staining, cells were fixed with 4% paraformaldehyde in PBS at room temperature for 15 min, and incubated with HaloTag® SaraFluor 650 T Ligand (1 μM) at room temperature for 30 min.

## Fixed-cell imaging with confocal microscopy

Cells were seeded on coverslips (13 mm No.1 S, MATSUNAMI) the day before fixation. Confocal microscopy was performed using a LSM880 with Airyscan (Zeiss) with a 63 × 1.4 Plan-Apochromat oil immersion lens or 100 × 1.46 alpha-Plan-Apochromat oil immersion lens. Images were analyzed and processed with Zeiss ZEN 2.3 SP1 FP3 (black, 64-bit) (ver. 14.0.21.201) and Fiji (ver. 2.0.0-rc-69/1.52p). Pearson's correlation coefficient was quantified by BIOP JACoP in Fiji plugin.

## Live-cell imaging with confocal microscopy

Live-cell imaging was performed using LSM880 with Airyscan (Zeiss) equipped with a 100 × 1.46 alpha-Plan-Apochromat oil immersion lens and Immersol™ 518 F/37 °C (444970-9010-000, Zeiss). The day before imaging, cells were seeded on a glass bottom dish (627870, Greiner bio-one) with growth medium without phenol red. During live-cell imaging, the dish was mounted in a chamber (STXG-WSKMX-SET, TOKAI HIT) to maintain the incubation conditions at 37 °C and 5% CO$_2$.

Images were acquired at intervals of 6 s, analyzed and Airyscan processed with Zeiss ZEN 2.3 SP1 FP3 (black, 64 bit) (ver. 14.0.21.201) and Fiji (ver. 2.0.0-rc-69/1.52p). For HaloTag7 staining before live-cell imaging, cells were cultured in medium containing HaloTag7 ligand at 37 °C for 30 min in a 5% $CO_2$ incubator.

## In vitro assay of recombinant TBK1-dependent phosphorylation of STING

Cells were collected in an ice-cold buffer (50 mM Tris–HCl pH 7.4, 100 mM NaCl, 1 mM EGTA, 2 mM DTT, 200 mM sucrose) containing protease inhibitors (25955-11, nacalai tesque), and phosphatase inhibitors (8 mM NaF, 12 mM β-glycerophosphate, 1 mM $Na_3VO_4$, 1.2 mM $Na_2MoO_4$, 5 μM cantharidin, and 2 mM imidazole), homogenized with 2 passages through a 27-gauge needle after 6 passages through a 23-gauge needle and centrifuged at $3000 \times g$ for 5 min at 4 °C. The post-nuclear supernatant was overlaid on 10 μL of 2 M sucrose and centrifuged at $100,000 \times g$ for 1 h at 4 °C. The membrane fractions were resuspended in a buffer (50 mM Tris–HCl pH 7.4, 100 mM NaCl, 1 mM EGTA, 2 mM DTT, 200 mM sucrose, 20 mM $MgCl_2$, protease inhibitors, and phosphatase inhibitors), and incubated with ATP (1 mM) and recombinant TBK1 (100 ng) at 37 °C for 30 min. The samples were then subjected to SDS-PAGE and phosphorylation of STING was detected by western blot.

## Liposome co-sedimentation assay

Recombinant mNeonGreen-iD4H protein was expressed in *E. coli* and purified with Ni-NTA (Qiagen) according to the manufacturer's protocol[67]. Lipid mixtures (Dioleoyl phosphatidylcholine and cholesterol) were dried under nitrogen gas and hydrated in buffer A (20 mM HEPES-NaOH (pH 7.4), 100 mM sucrose, 100 mM KCl, and 1 mM EDTA) for 15 min at room temperature, and vortexed briefly. mNeonGreen-iD4H recombinant protein was diluted with HEPES Buffered Saline, incubated with the liposomes in buffer A for 30 min at 30 °C. The mixture was centrifuged at 20,000 x g for 30 min at 25 °C. The resultant supernatant was collected, and the pellet was washed twice with buffer A. The pellet was then subjected to SDS-PAGE and Coomassie Brilliant Blue (CBB) Staining.

## Measurement of lipid order using di-4-ANEPPDHQ

Cells were incubated with di-4-ANEPPDHQ (1 μg mL$^{-1}$) in DMEM at 37 °C for 30 min followed by image acquisition. All images for di-4-ANEPPDHQ were acquired on LSM880 (Zeiss) with a 488 nm Ar laser and a 633 nm He-Ne laser line using a $63 \times 1.4$ Plan-Apochromat oil immersion lens. di-4-ANEPPDHQ was excited at 488 nm, and the emissions were captured at the following manual bandwidth settings of the spectral detection channel: Ch-L, 505–530 nm; and Ch-H, 675–700 nm. The SaraFluor™ 650 T was excited at 633 nm and detected with the spectral detector at 670–750 nm. di-4-ANEPPDHQ intensity images were converted into generalized polarization (GP) images[41], with each pixel calculated in Fiji from the two di-4-ANEPPDHQ intensity images according to the equation: GP = $(I_{Ch-L} - I_{Ch-H}) / (I_{Ch-L} + I_{Ch-H})$. GP values in the Golgi area or TBK1 foci were obtained by using binarized images of HaloTag7-Rab6 or TBK1-HaloTag7 generated by Trainable Weka Segmentation, a machine learning tool for microscopy pixel classification in Fiji plugin. GP values in the cytoplasm without plasma membrane were obtained by measuring the mean of GP values in a region 2 μm inside the cell edge.

## qRT-PCR

Total RNA was reverse-transcribed by using ReverTraAce qPCR RT Master Mix with gDNA Remover (TOYOBO). Quantitative real-time PCR (qRT-PCR) was performed using KOD SYBR qPCR (TOYOBO) and LightCycler 96 (Roche). The sequences of the primers were as follows. 5′-AGTGCTGCCGTCATTTTCTGCCTC-3′ (mouse Cxcl10; sense primer) and 5′-GCAGGATAGGCTCGCAGGGATGATT-3′ (mouse Cxcl10;

antisense primer); 5′-AGGTCGGTGTGAACGGATTTG-3′ (mouse Gapdh; sense primer) and 5′-TGTAGACCATGTAGTTGAGGTCA-3′ (mouse Gapdh; antisense primer). Target gene expression was normalized based on Gapdh content.

## Lipid analysis

Cells were collected in an ice-cold buffer (50 mM Tris–HCl pH 7.4, 100 mM NaCl, 1 mM EGTA, 2 mM DTT, 200 mM sucrose) containing protease inhibitors (25955-11, nacalai tesque), and phosphatase inhibitors (8 mM NaF, 12 mM β-glycerophosphate, 1 mM $Na_3VO_4$, 1.2 mM $Na_2MoO_4$, 5 μM cantharidin, and 2 mM imidazole), homogenized with 2 passages through a 27-gauge needle after 6 passages through a 23-gauge needle, and centrifuged at $3000 \times g$ for 5 min at 4 °C. The post-nuclear supernatant was overlaid on 10 μL of 2 M sucrose and centrifuged at $100,000 \times g$ for 1 h at 4 °C. Total lipids were extracted by the Bligh and Dyer method[68]. To detect cholesterol, lipid extracts were subjected to high-performance TLC (Merck) separation with hexane/diethyl ether/ acetic acid (80:20:1, vol:vol:vol). Cholesterols were stained with a mixture of ferric chloride/sulfuric acid/acetic acid by heating[69].

## RNA interference

siRNA specific to Ch25h (Ch25h Stealth Select RNAi) purchased from Thermo Fisher Scientific. Negative control siRNA was purchased from Dharmacon. A total of 5 nM siRNA was introduced to cells using Lipofectamine RNAiMAX (Invitrogen) according to the manufacturer's instruction. Cells were further incubated for 72 h for subsequent experiments.

## Cholesterol add-back experiments

Cells were incubated with medium containing cholesterol-methyl-β-cyclodextrin complex (Chol/MβCD) (3.9 mM) for 3 h.

## Statistical analysis

Error bars displayed in bar plots throughout this study represent s.e.m. unless otherwise indicated and were calculated from triplicate or quadruplicate samples. In box-and-whisker plots, the box bounds the interquartile range (IQR) divided by the median, and whiskers extend to a maximum of $1.5 \times IQR$ beyond the box. The corresponding data points are overlayed on the plots. The data were statistically analyzed by performing one-way ANOVA followed by Tukey-Kramer *post hoc* test with KNIME (4.6.3) and R (4.0.2).

The number of STING molecules per cluster and the number of STING molecules within 100 nm radius around TBK1 spots were statistically analyzed by performing Welch's *t*-test (both-sided). In cases requiring multiple statistical tests, the significance level was corrected by the Holm-Sidak method.

When the histogram of colocalization durations was fitted with a single exponential decay function, the 68.3% confidence limit was given as the fitting error for the decay time. The statistical analysis for these distributions was performed using log-rank test (statistical survival analysis). The significance level was corrected by the Holm-Sidak method.

## Data acquisition of live-cell PALM of mEos4b-STING or mEos4b-cavin1

For live-cell PALM observation of mEos4b-STING, MEFs were sparsely seeded in a glass-base dish ($4 \times 10^3$ cells on the glass window of 12 mm in diameter, 0.15-mm-thick glass; Iwaki), and grown in DMEM without phenol red supplemented with 10% FBS for 2 days before each experiment. The data acquisitions of PALM of mEos4b-STING were performed at 37 °C and at a video rate (33 ms/frame) or 5 ms/frame for 5400 frames with an image size of $1024 \times 1024$ pixels or $512 \times 512$ pixels, respectively, using an Olympus IX-83 inverted microscope ($100 \times 1.50$ NA oil objective) equipped with a high-speed gate image

intensifier (C9016-02MLG; Hamamatsu Photonics) coupled to an sCMOS camera (OCRA-Flash4.0 V2; Hamamatsu Photonics). mEos4b-STING molecules in cells were illuminated with the oblique-angle illumination mode using an activation laser (405 nm, approximately 4 nW/μm² and 20 nW/μm² for observation at 33 ms/frame and 5 ms/frame, respectively) and an excitation laser (561 nm, approximately 7 μW/μm² and 14 μW/μm² for observation at 33 ms/frame and 5 ms/frame, respectively). The single-molecule localization precision of mEos4b was 20 ± 1 nm. The final magnifications were 133×, resulting in pixel sizes of 47.1 nm (square pixels).

For live-cell PALM observation of mEos4b-cavin1, PC3 cells were transfected with cDNA encoding mEos4b-cavin1 by 4D-nucleofector (LONZA) and the cells were sparsely seeded in a glass-base dish, and grown in HAM/F12 with 10% FBS for 2 days before observation. The data acquisitions of PALM of mEos4b-cavin1 were performed at 37 °C and at a video rate (33 ms/frame) with an image size of 1024 × 1024 pixels. Single mEos4b-cavin1 molecules in cells were observed by total internal reflection microscopy.

## In vitro blinking observation of immobilized mEos4b and analysis

For correction of multiple blinking of mEos4b, we observed in vitro blinking and photoactivation behaviors of purified mEos4b. mEos4b gene was cloned in the pET15b plasmid, transformed in *E coli*, BL21 (DE3) to overexpress proteins, and mEos4b was purified using a His60 Ni Superflow Cartridge Purification Kit (Clontech), followed by a gel-filtration step using a PD-10 column (Cytiva). The purified mEos4b was diluted to 5 nM in a solution of 1% polyvinyl alcohol (PVA, nacalai tesque) in HBSS (pH = 7.4). Then, 150 μl of the mEos4b solution was placed in a glass window of the glass base dish and was incubated for 1 h at room temperature to form a uniform thin layer. After washing with HBSS, 1 ml HBSS was added to the dish.

The blinking and photoactivation behaviors of mEos4b molecules were observed under the same laser illumination condition as that in cells. As shown in Fig. 1b, we observed single mEos4b molecule emission trace, and measured the number of times mEos4b blinks $<N_{blink}>$ before a photobleaching event, the fluorescence-on time ($T_{on}$) before the molecule goes into the dark state or it photobleaches, the fluorescence-off time ($T_{off}$) or the time the molecule spends in the dark state as previously reported[27]. The estimation of these parameters from the obtained images of mEos4b immobilized in PVA was performed by using the GDSC SMLM plugin of ImageJ (Time threshold was set to 2.5 s). The histogram of relationship between $<N_{blink}>$ and the probability was fitted by the predicted geometric distribution;

$$P(N_{blink} = n) = h^n(1-h) \tag{1}$$

where $h = k_d/(k_d+k_b)$ is the probability of transition to the dark state and $k_d$ and $k_b$ are the rates of transition to the dark and the photobleach states, respectively (Fig. 1d). The histogram of relationship between $T_{on}$ and probability density function (PDF) was fitted with single exponential decay function, providing $k_b + k_d$ (Fig. 1e). $N_{blink}$ (= $k_d/k_b$) was estimated from these values.

## Analysis of images obtained by super-resolution microscopy

The detection of the fluorescent spots in the images was performed by using the ThunderSTORM plugin of ImageJ with "Wavelet filtering" (B-Spline order = 6 and B-Spline scale = 6.0) and the "Local maximum method" (Peak intensity threshold = 30−50 and Connectivity = 8-neighborhood). After spot detection, the post-processing steps of "Remove duplications" (Distance threshold = uncertainty) and "Drift correction" (cross-correlation with 5 bins) were further performed.

Several methods for PALM image segmentation have been proposed, but which method is the most appropriate is under debate and depends on characteristics of the objects of observation such as molecular density, cluster size, and the ratio of molecules in clusters to those in bulk phase, etc[32]. For example, a pair-correlation method identifies molecular interactions inside clusters by analyzing distance correlation between localizations, but this method does not give us the number and position of clusters, and can be applied only to small clusters of relatively homogenous sizes. Density based spatial clustering analysis with noise (DBSCAN), which allows the classification of molecules in clusters, is known to be sensitive to background noise and difficult to parameterize experimentally. In the present study, we employed two kinds of ways for the image segmentation to detect mEos4b-STING clusters, kernel density estimation (KDE) and SR-Tesseler. In these methods, image segmentation is performed directly from the localization coordinates, which is applicable for the analysis of clusters of heterogenous size, and insensitive to background noise. Furthermore, these methods can be applicable to a wide range of the objects of observation[32]. SR-Tesseler[34] is one of the most frequently used methods for PALM image segmentation. Meanwhile, KDE[31,70] is a traditional image segmentation method and can rapidly determine the criterion for segmenting PLAM images as reported previously[71] and described below. KDE provides a way to interpolate object boundaries without bias by the random fluctuations of activated boundary fluorophores. In this study, the binarized PALM images of STING clusters were used for estimation of the residency time of single-molecules of TBK1-HaloTag7 inside the STING clusters upon stimulation (Fig. 8d, e).

The datasets obtained by Thunderstorm were imported to the KDE software (MATLAB) for the cluster analysis and the image reconstruction. We used not only localization coordinates, but also uncertainty outputted by Thunderstorm, which represents the degree of spatial precision of the spot. Let $x_i, y_i$, and $u_i, i = 1,2,\cdots,N$ be the horizontal and vertical localization coordinates and uncertainty of spot $i$, respectively. $N$ represents the total number of spots. Considering the uncertainty of the location measurement of spot $i$, it is considered that the existence probability $p_i(x,y)$ of the spot $i$ spreads to the Gaussian distribution with the center coordinate $x_i,y_i$ and the standard deviation (S.D.) that is proportional to the uncertainty $u_i$ as follows:

$$p_i(x,y) = \frac{1}{2\pi(Au_i)^2} e^{-\frac{(x-x_i)^2+(y-y_i)^2}{2(Au_i)^2}} \tag{2}$$

in which, $A$ is the proportional coefficient of S.D. to the uncertainty $u_i$; that is estimated as $A \approx 6$ by some preliminary experiments. Gaussian distribution constructing each existence probability $p_i(x,y)$ is called Gaussian kernel. The existence probability distribution for all spots, i.e., the PALM image is obtained by the average of the existence probability distribution $p_i(x,y)$ as

$$p(x,y) = \frac{1}{N}\sum_{i=1}^{N} p_i(x,y) \tag{3}$$

excluding spots having extremely small or large uncertainty $u_i$ because such spots are likely to be noise.

It can be considered that clusters exist where the PALM image, which represents the cluster existence probability distribution, takes above a certain threshold value $\theta$ because the spots appear uniformly inside the clusters with a certain probability. It is necessary to appropriately determine the optimal threshold value $\theta$ for each obtained PALM image because the optimal value differs depending on the target molecule and the experimental environments. In general, the histogram of $p(x,y)$ values for all $x,y$ forms a bimodal shape consisting of both larger values created by dense spots appeared in clusters and smaller values created by sporadic noise. Therefore, the threshold value $\theta$ should be determined so as to separate these two clusters. Otsu's method is well known as a method for determining such a threshold value[72]. Let $S_{in}$ and $S_{out}$ be sets of coordinates $(x,y)$ where $p(x,y) \geq \theta$ and $p(x,y) < \theta$, respectively. The sets $S_{in}$ and $S_{out}$ mean the

inside and outside of clusters, respectively. The number of elements of the sets $S_{in}$ and $S_{out}$ are respectively represented by $N_{in}$ and $N_{out}$. The intra-class variances of the sets $S_{in}$ and $S_{out}$ are calculated by

$$\sigma^2[S_{in}] = \frac{1}{N_{in}} \sum_{(x,y) \in S_{in}} p(x,y) \tag{4}$$

$$\sigma^2[S_{out}] = \frac{1}{N_{out}} \sum_{(x,y) \in S_{out}} p(x,y) \tag{5}$$

The average of these intra-class variances is

$$\sigma^2[S_{in}, S_{out}] = \frac{N_{in}\sigma^2[S_{in}] + N_{out}\sigma^2[S_{out}]}{N_{in} + N_{out}} \tag{6}$$

The Otsu's method determines the optimal threshold value $\hat{\theta}$ so as to minimize the average of the intra-class variances $\sigma^2[S_{in}, S_{out}]$ as follows:

$$\hat{\theta} = \arg\min_{\theta} \sigma^2[S_{in}, S_{out}] \tag{7}$$

In this study, the clusters are determined using the above Otsu's method.

Alternatively, the dataset obtained by Thunderstorm was also imported to the SR-Tesseler software for the cluster analysis and image reconstruction[34]. Cluster contour detection was performed by using a Voronoi polygon density factor of 2 without correction of blinking and multi-ON frame detections by the"Detection cleaner" function (within the whole ROI).

Multiple blinking effects were corrected by two methods. One of them is the most commonly used method to merge events that appear close in space and time which is referred to as dark-time thresholding (DTT)[27]. To estimate the numbers of mEos4b-STING in clusters, the numbers of localizations in clusters obtained by the KDE or SR-Tesseler method were divided by $1 + <N_{blink}>$, which is the number of localizations per one mEos4b molecule (2.7 for observation at 33 ms/frame and 2.9 for observation at 5 ms/frame observation). The previous study demonstrated[27] that this method accomplishes unbiased counting of molecules in clusters.

DTT has been most commonly used for the correction of multiple-blinking, but we need to assume that the blinking behaviors of isolated mEos4b are maintained also in cells. Therefore, to validate the STING numbers per cluster, we also corrected multiple-blinking artifacts by another method, using the PALM dataset and the parameters of a realistic model of fluorescent protein photophysics, which was recently reported and referred to as model-based correction (MBC)[35]. MBC requires neither user input nor additional calibration data.

Using two methods of PALM image segmentation (KDE and SR-Tesseler) and two methods of multiple-blinking correction (division by $1 + <N_{blink}>$ and MBC), we estimated averaged numbers of mEos4b-STING in clusters by three ways. Namely, we performed PALM image segmentation and multiple-blinking correction in the following three ways; (1) The number of localizations segmented by KDE was divided by $1 + <N_{blink}>$, (2) The number of localizations segmented by SR-Tesseller was divided by $1 + <N_{blink}>$, (3) mEos4b-STING spots corrected by MBC was segmented by SR-Tessler.

## Simultaneous dual-color observation of live-cell PALM of mEos4b-STING and dSTORM of GM130-HaloTag7, TGN38-Halo-Tag7, TfnR-HaloTag7 or HaloTag7-Rab11 and coordinate-based colocalization analysis of PALM and dSTORM data

Simultaneous data acquisitions of live-cell PALM of mEos4b-STING and dSTORM of GM130-HaloTag7 (a CGN protein), TGN38-HaloTag7 (a

TGN protein), TfnR-HaloTag7 (an endosome protein) or HaloTag7-Rab11 (an endosome protein) were performed at 5 ms resolution and at 37 °C. The data acquisition of PALM of mEos4b-STING was performed as described in the method section of "Data acquisition of live-cell PALM of mEos4b-STING". For live-cell dSTORM of GM130-HaloTag7, TGN38-HaloTag7, TfnR-HaloTag7 or HaloTag7-Rab11, these molecules expressed in cells and labeled with SaraFluor650B (SF650B) were illuminated with the oblique-angle illumination mode using an excitation laser (647 nm, approximately 16 $\mu$W/$\mu$m$^2$). In the excitation arm, a multiple-band mirror (ZT405/488/561/647rpc, Chroma) was employed. The two-color fluorescence images of mEos4b and SF650B were separated into the two detection arms of the microscope by a dichroic mirror (Chroma: ZT561rdc-xr-UF2 or ZT647rdc-UF2). The detection arms were equipped with band-pass filters of FF01-600/37-25 (Semrock) or ET700/75 (Chroma), and the data acquisitions of PALM/dSTROM were performed at 5 ms/frame with an image size of 512 × 512 pixels, using two high-speed gate image intensifiers (C9016-02MLG; Hamamatsu Photonics) coupled to two sCMOS cameras (OCRA-Flash4.0 V2; Hamamatsu Photonics). The superimposition of images in different colors obtained by two separate cameras was performed as reported previously[73].

To quantitatively analyze the degree of colocalization between mEos4b-STING and GM130-HaloTag7-SF650B, TGN38-HaloTag7-SF650B, TfnR-HaloTag7-SF650B, or SF650B-HaloTag7-Rab11, we performed coordinate-based colocalization (CBC) analysis of PLAM and dSTORM data according to ref. 36. with modification. Briefly, to estimate CBC values, for each molecule of protein A, the number of localizations of protein A (mEos4b-STING) and protein B (GM130-HaloTag7-SF650B, TGN38-HaloTag7-SF650B, TfnR-HaloTag7-SF650B, or SF650B-HaloTag7-Rab11) within circles of the increasing radius was calculated, respectively, providing the density gradients of localizations of protein A and protein B around this molecule.

$$D_{Ai,A}(r) = \frac{N_{Ai,A}(r)}{\pi r^2} \times \frac{\pi R_{max}^2}{N_{Ai,A}(R_{max})} = \frac{N_{Ai,A}(r)}{N_{Ai,A}(R_{max})} \times \frac{R_{max}^2}{r^2} \tag{8}$$

$$D_{Ai,B}(r) = \frac{N_{Ai,B}(r)}{N_{Ai,B}(R_{max})} \times \frac{R_{max}^2}{r^2} \tag{9}$$

Here, $N_{Ai,A}(r)$ is the number of localization of protein A within the distance r around protein A$i$, and $N_{Ai,B}(r)$ is the number of localization of protein B within the distance r around A$i$. Then, these density gradients were corrected for the area ($\pi r^2$), normalized by the number of localizations within the largest observed distance $R_{max}$ and divided by the largest area for protein A ($\frac{N_{Ai,A}(R_{max})}{\pi R_{max}^2}$) and protein B ($\frac{N_{Ai,B}(R_{max})}{\pi R_{max}^2}$). Namely, the density gradients were corrected by the density at the maximum radius respectively for protein A and protein B. $R_{max}$ and dR (the bin of radius for analysis) were set at 500 nm and 50 nm, respectively, and if both the number of localizations of protein A and that of protein B were less than 50 ($= \frac{R_{max}}{dR} \times 5$) in a circle with $R_{max}$, CBC values were not calculated because the density gradients cannot be accurately calculated. A uniform distribution gives an expected value of $D(r) = 1$ for all r.

The two distributions were compared by calculating a rank correlation coefficient (Spearman), in which the colocalization coefficient was weighted by a value proportional to the distance to the nearest neighbor to avoid long-distance effects[36].

$$S_{Ai} = \frac{\sum_{r_j=0}^{R_{max}}(O_{D_{Ai,A}}(r_j) - \bar{O}_{D_{Ai,A}})(O_{D_{Ai,B}}(r_j) - \bar{O}_{D_{Ai,B}})}{\sqrt{\sum_{r_j=0}^{R_{max}}(O_{D_{Ai,A}}(r_j) - \bar{O}_{D_{Ai,A}})^2}\sqrt{\sum_{r_j=0}^{R_{max}}(O_{D_{Ai,B}}(r_j) - \bar{O}_{D_{Ai,B}})^2}} \tag{10}$$

Here, $\overline{O_{D_{Ai,A}}}(r_j)$ is the rank of $O_{D_{Ai,A}}(r_j)$ calculated after Spearman, and $\bar{O}_{D_{Ai,A}}$ is the arithmetic average of $O_{D_{Ai,A}}(r_j)$. The colocalization value $C_{Ai}$, was calculated as $C_{Ai} = S_{Ai} \times e^{-\left(\frac{E_{Ai,B}}{R_{max}}\right)}$, with $E_{Ai,B}$ as the distance from $Ai$ to the nearest neighbor from protein B. $C_{Ai}$ (CBC score) was calculated for every single-molecule localization and ranged from −1 (anti-colocalized or segregated), through 0 (no colocalization), to +1 (totally colocalized). Furthermore, $C_{Bi}$ was also calculated as well. The summation of $C_{Ai}$ and $C_{Bi}$ in the bins between 0.7 and 1 was used as an index for the colocalization. For control analysis, dSTORM images of GM130-HaloTag7-SF650B, TGN38-HaloTag7-SF650B, TfnR-HaloTag7-SF650B, or SF650B-HaloTag7-Rab11 were rotated by 180° and overlapped with PALM images of mEos4b-STING and the CBC scores were estimated.

**Simultaneous live-cell PALM of mEos4b-STING, to visualize STING clusters and single-molecule imaging of TBK1-HaloTag7, to track their recruitment to STING clusters, and estimation of the residency lifetimes of TBK1-HaloTag7 inside the STING cluster**

Simultaneous data acquisitions of live-cell PALM of mEos4b-STING and single-molecule tracking of TBK1-HaloTag7 labeled with Sara-Fluor650T (SF650T) were performed at 37 °C. The data acquisition of PALM of mEos4b-STING was performed as described in the method section of "Data acquisition of live-cell PALM of mEos4b-STING".

Single-molecules of SF650T bound to TBK1-HaloTag7 were illuminated with the oblique-angle illumination mode using a 647 nm laser at 2.0 μW/μm². The two-color fluorescence images of mEos4b and SF650T were recorded at 33 ms/frame. Under the conditions, the photobleaching lifetimes for SF650T was 7.3 ± 0.1 s.

Each individual fluorescent spot in the image was identified and tracked as described previously[74,75]. The superimposition of images in different colors obtained by two separate cameras was performed as reported previously[73]. The PALM image segmentation of mEos4b-STING was performed by KDE and the boundaries of STING clusters were determined by binarizing the STING cluster image as described in the section of "Analysis of images obtained by super-resolution microscopy". After binarization, the coordinates of the pixels located at the edges of the STING clusters were defined as the boundary. We measured the periods of trajectories of TBK1-HaloTag7 labeled with SF650T inside the boundary of STING clusters, and estimated the residency lifetimes by fitting the histograms with a single-exponential decay function. For control analysis to measure the lifetimes of non-specific colocalization, PALM images of mEos4b-STING after DMXAA stimulation were superimposed with images of single-molecules of TBK1-HaloTag7-SF650T before the stimulation.

To quantitatively evaluate STING densities in the cluster at which TBK1-HaloTag7-SF650T molecules were recruited, we estimated the numbers of mEos4b-STING molecules within 100 nm radius around all the recruited TBK1-HaloTag7-SF650T spots for all the frames. For control analysis, we generated the pseudo-trajectories by shifting the trajectories of TBK1-HaloTag7-SF650T in random directions by random distances. In PALM image, let $x(t)$ and $y(t)$ be the x and y coordinates in the PALM image is defined as $x = 1, 2, \cdots, N_x$ and $y = 1, 2, \cdots, N_y$, respectively. We generated two random natural numbers $x_0$ and $y_0$, both less than or equal to $N_x$ and $N_y$, respectively, then use $x(t) - x(0) + x_0$ and $y(t) - y(0) + y_0$ as the trajectories of the control. In case that any part of the trajectories exceeds the range of x and y coordinates in the PALM image, we regenerate the random numbers and start over the process. Then, we estimated the numbers of mEos4b-STING molecules within 100 nm around the same numbers of randomized TBK1-HaloTag7-SF650T spots in silico. Subsequently, we created the histograms of the difference between the numbers of STING molecules around the observed TBK1 spots and the randomized spots in silico, which was normalized by periods and areas of the observations (Fig. 8f,

bottom). The reason for choosing 100 nm as the distance around the recruited TBK1 molecules to quantify the number of STING molecules is that shorter distances make analysis more difficult. For example, if we employ 50 nm as the distance, the area the number of STING molecules become one forth and the bars in the histogram of Fig. 8f fluctuate very much and the negative values often appear. It becomes hard to perform the same analysis shown in Fig. 8f.

## Reporting summary
Further information on research design is available in the Nature Portfolio Reporting Summary linked to this article.

## Data availability
The data sets generated during and/or analyzed during the current study are available from the corresponding author upon request. Source data are provided with this paper.

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

## Acknowledgements

This work was supported by JSPS KAKENHI Grant Numbers JP19H00974 (T.T.), JP21H02424 (K.G.N.S.), JP20K21387 (K.G.N.S.), JP20H05307 (K.M.), JP20H03202 (K.M.), JP19K06536 (K.Tanaka.), JP21K06076 (T.K.), JP22K11920 (Y.Y.), JSPS Research Fellowship for Young Scientists (K.Takahashi., H.K. and Y.K.), AMED-PRIME (17939604) (T.T.), JST CREST (JPMJCR18H2) (K.G.N.S.), JST CREST (JPMJCR21E4) (K.M.), the establishment of university fellowships towards the creation of science technology innovation, Grant Number JPMJFS2102 (H.K.), the Subsidy for Interdisciplinary Study and Research concerning COVID-19 (Mitsubishi Foundation) (T.T.), National Cancer Center Research and Development Fund (2023-A-03) (K.G.N.S.), Takeda Science Foundation (T.T., K.G.N.S., and K.M.), The Uehara Memorial Foundation (K.G.N.S.), Mizutani Foundation for Glycoscience (K.G.N.S.), Daiichi Sankyo Foundation of Life Science (K.G.N.S.), Research Foundation for Opto-Science and Technology (K.G.N.S.), The Naito Foundation (K.G.N.S.), Grant for Basic Science Research Projects from the Sumitomo Foundation (K.M.), SGH Cancer Research Grant (K.M.), and Research Grant of the Princess Takamatsu Cancer Research Fund (K.M.). T.M. was supported by Nagoya University CIBoG program from MEXT WISE program.

## Author contributions

H.K. and K.Takahashi. designed and performed the experiments, analyzed the data, and interpreted the results; K.M. designed the experiments, analyzed the data, interpreted the results, and wrote the paper; T.M. purified mEos4b and performed in vitro observation; K.M.H. and Y.Y. developed analysis software; F.K. performed the experiments with 25-HC; Y.U. designed the experiments, analyzed the data, and interpreted the results with OSW-1; Y.K. discussed the results; Y.N. and M.S. prepared Tag-free recombinant TBK1; T.K. and K.Tanaka. designed and performed the experiments, analyzed the data, and interpreted the results with iD4H and di-4-ANEPPDHQ; H.A. discussed the results; K.G.N.S. designed and performed the experiments, analyzed the data, and interpreted the results, and wrote the paper; T.T. designed the experiments, interpreted the results, and wrote the paper.

## Competing interests

M.S. is a Chief Scientific Officer of Carna Biosciences, Inc. and owns stocks of Carna Biosciences, Inc. T.T. receives research funding from Carna Biosciences, Inc. All the other authors have no competing/conflict of interest.
