## [Peer Review File · Nature Communications]

Single-molecule localization microscopy reveals STING clustering through palmitoylation-dependent accumulation of cholesterol in the TGNREVIEWER COMMENTS

Reviewer #1 (Remarks to the Author):

In this study, the Tomohiko Taguchi group identifies the activation of STING requires cholesterol at the TGN for aggregation. And they utilize single-molecule imaging to demonstrate that STING becomes mega-clustered at the trans-Golgi network. With this technology, the aggregation of STING (in this paper called mega-cluster of STING) was monitored with the treatment of a palmitoylation inhibitor. Moreover, the recruitment of TBK1 by STING aggregation was directly observed using this technology. The authors got the same conclusion from this technology as the previous studies. The novel information the authors get from this study is that cholesterol is another layer of regulation of STING clustering and activation. Overall, the authors provide a cutting-edge technology to characterize the activation of STING and some novel information on STING activation regulation. The issues are listed below.

1. Does the STING cluster specifically emerge in TGN? Given that the TGN marker, e.g. TGN38/46, also exist in scattered endosomes, the author should test whether the STING cluster also locates in endosomes (may test by EEA1) or other related organelle.
2. In figure 4 J K M, the cholesterol level seems to be induced after STING stimulation, what is the possible reason? Is iD4H only probe for cholesterol detection? Other probes should be used to confirm this observation.
3. In figure S1A, the expression level of STING in STING KO MEFs is inconsistent with p-TBK1 and p-STING.
4. Please repeat this experiment in Fig 2A, 2B, 2G, 3B, 3I, S7A and S10B, total protein level of IRF3, TBK1, STING and tubulin are not in a similar level.
5. The authors should reconsider the statement in page 7, line 9. Why discuss cGAS and cGAMP? There is no connection with the theme. If you want to exclude cGAS and cGAMP, cGAS KO cells should be included.
6. In fig 3A and 3B, p-STING and p-IRF3 seemed to be inhibited obviously, but p-TBK1 was not affected dramatically, please explain.
7. Page 8, line 10, please add a control without TBK1. In fig 3I, cholesterol was essential for priming STING for its phosphorylation, but how can you examine whether STING is primed for the phosphorylation by TBK1.

8. Page 9, line 6, please explain why you test the signal of mNeonGreen-iD4H at the perinuclear region. Does cholesterol traffic with STING to endosome?
9. Fig S1B and S8A, the mEos4b-STING clusters colocalized with GM130 and TGN38 is different in Fig S1B. It is not consistent.
10. OSBP should use the full name when it is first mentioned.

Reviewer #2 (Remarks to the Author):

In the manuscript entitled “Activation of STING requires cholesterol-mediated mega-clustering at the TGN”, Kemmoku et. al. investigate the regulation of STING activation – a crucial step in the cGAS-STING signaling pathway which is a key mediator of inflammation in the context of infection, cellular stress and tissue damage. The authors employ quantitative photoactivated localization microscopy (PALM) of reconstituted mEos4b-STING in MEF cells to measure cluster formation of STING. Co-staining with cis- and trans-Golgi markers revealed a time-dependent appearance of large STING clusters at the trans-Golgi network after treatment of the cells with the STING agonist DMXAA. The authors demonstrate a correlation between the average size of STING clusters and the degree of downstream pathway activation. This is in agreement with their observation that STING clusters facilitate the recruitment of the downstream signaling molecule TBK1 and presumably its activation at STING clusters. Using palmitoylation inhibitors as well as a palmitoylation-deficient mutant of STING, the authors found a dependency between palmitoylation of STING and cluster formation. Furthermore, the authors show that STING clusters formation and the degree of downstream pathway activation is modulated by the presence of cholesterol at the Golgi membranes and that STING clusters localize to areas of increased membrane order. Based on these findings the authors propose a model of palmitoylation and cholesterol-dependent STING clustering in membrane microdomains at the trans-Golgi network that has a direct correlation with the degree of activation of the signaling pathway. The study is very interesting for the field in two ways, because it succeeds at linking oligomerization measured directly in cells with function, and because it advances our understanding of the molecular mechanisms in the STING pathway.

Major comments:

- For quantification of STING clusters, the authors apply a previously described kinetic model for the quantification of PALM data. The accuracy of the kinetic model relies on the accuracy of the calibration and the correct determination of the photophysical properties of the used fluorophore (Nblink, Ton, Toff, etc). Here, the authors used PVA-embedded mEos4b monomers on glass for calibration. The physical and chemical environment of a fluorophore or fluorescent protein might greatly influence its photophysical properties (transition rates from fluorescent to dark state, blinking rates, photobleaching, etc). It is therefore of essential importance to account for differences of these properties in the calibration sample compared to the actual measurement sample (i.e. mEos4b in PVA on glass for calibration vs mEos4b tagged to STING in cells).

It is important to discuss the possible influence of PVA embedding on the fluorescent properties in the text/methods as a limitation or to compare the measured calibration values to the ones obtained from a calibration with different sample preparation (e.g. biotinylated mEos4b on streptavidin-coated glass coverslips as in the original publication (Lee et. al., PNAS, 2012)).

- Additionally, for accurate calibration the authors should show only monomeric mEos4b is measured. It would be helpful to show a FWHM distribution or brightness distribution of the measured molecules to demonstrate that mEos4b was monomeric.

- However, the question whether oligomerization into clusters of hundreds of molecules changes the photophysical properties of mEos4b compared to the monomeric form should be addressed as it would also have a strong impact on the quantification of the oligomeric state. More adequate calibration references using oligomers of mEos4b of known stoichiometry that are closer to the oligomer size investigated here would be desirable.

- The size of STING clusters at the TGN might be partially determined by the abundance of Golgi membrane and the STING expression levels (especially for late time points after activation). It would be interesting to check for this correlation by normalizing the size of STING clusters to a TGN signal and the absolute cellular abundance of STING.

- The dependence of STING signaling on the presence of cholesterol at the Golgi is

demonstrated by a reduction of TBK1-phosphorylation in cells, a decrease in downstream mRNA expression in vivo and a reduction of the number of STING clusters upon OSBP inhibition (Fig. 3 A to G). It is further supported by an in vitro cholesterol reconstitution assay (Fig. 3 H, I). It is highly important to control the effect of cholesterol depletion at the Golgi (after OSW-1 or 25-HC treatment) on overall Golgi size, membrane abundance and membrane integrity at the Golgi as well as its overall 'functionality'.

- Despite the reduction in the pathway activation and STING cluster number and size under OSW 1 treatment conditions (Fig. 3 A, B), in Fig. 3 J, K and M it is evident that both, STING and TBK1 still cluster at perinuclear (Golgi) regions under these conditions, indicating that cholesterol might influence but is not required for their clustering. The authors should discuss this observation in more detail and avoid describing the role of cholesterol as 'essential'. Furthermore, it is important to control that the reduction in clustering is not consequence of overall reduced Golgi membrane abundance after cholesterol depletion as mentioned above.

- The authors show that the staining with mNeonGreen-iD4H can be used to detect abundances of cholesterol of higher than 20 % (Fig. S8 D). Treatment with DMXAA for 60 min leads the detection of mNeonGreen-iD4H at Golgi regions (Rab6-positive membranes, Fig. 3 M) indicating that the induction of STING signaling using DMXAA triggers the enrichment of cholesterol at the Golgi. The authors should test or discuss if this is caused by enhanced cholesterol transport to Golgi membranes (e.g. through the secretory pathway) or reorganization of Golgi cholesterol into enriched domains.

- The authors test for colocalization of mRuby3-STING, TBK1-mScarlet and Halo-Rab6 with mNeonGreen-iD4H (Fig. 3 J to N and Fig. S 8 E). In untreated samples (DMXAA 0 min) as well as after inhibition of cholesterol transport to Golgi membranes (OSW-1 treatment), hardly any mNeonGreen-iD4H signal is detectable. Without decent signal intensity, it is not possible to measure colocalization or to calculate trustable Pearson's correlation coefficients. Furthermore, the mNeonGreen-iD4H signal in the representative images of DMXAA-treated samples seems saturated which impairs the accuracy of the analysis as well. The authors should improve this analysis in order to make trustable conclusions or should

quantify the signal intensity of mNeonGreen-iD4H as an indicator for the cholesterol abundance instead of colocalization analysis.

- Based on the influence of (i) the presence of cholesterol on Golgi membranes (Fig. 3), (ii) the palmitoylation of STING (Fig. 2) and (iii) the increase of lipid order/membrane rigidity on STING clustering and downstream pathway activation (Fig. S7), the authors propose in their model the requirement of the formation of lipid microdomains (“rafts”) for STING activation. While the phenomenon of “lipid rafts” is debated for decades as a hub for the enrichment and signaling of proteins, the author should be careful with their statement on the essentiality of lipid rafts for STING activation. In order to make this conclusion, it is essential to demonstrate that the formation of these lipid microdomains is a causative regulatory step for the pathway execution and to rule out that their formation is a consequence of it (i.e., the possibility that STING palmitoylation and its putative intrinsic property for clustering or oligomerization upon activation causing the reorganization of the membrane into a specialized micro-environment.

- In Fig. 4 A to D, the authors show the transient localization of TBK1 monomers to STING clusters upon DMXAA treatment with a colocalization duration of ~150 ms. How do the authors explain this transient localization and dissociation compared to the formation of large (stable) clusters of TBK1 as shown in Fig. 3 K?

- The authors demonstrate the increase of the number of STING clusters caused by disease-causative COPA variants in absence of the STING agonist DMXAA. This result indicates that COP-I-mediated retrograde transport might be involved in the reduction of STING clusters by removing STING from the CGN but doesn't provide evidence for the role of STING clustering on the outcome of the signaling pathway. The authors should provide data on the activation of the signaling pathway in presence of these COPA variants and demonstrate how STING is activated in absence of the STING agonist DMXAA or cGAMP under these conditions.

- The authors measured the number of STING molecules in a radius of 100 nm around TBK1-Halo spots and compared it to those around ‘randomized’ TBK1 spots in silico. This

methodology needs a much broader introduction and explanation. How are the TBK1 spots randomized? What are the parameters and assumptions? How is the localization of STING measured relative to the simulated TBK1 spots?

- The authors refer to “STING ‘mega’-clusters” several times throughout the manuscript. As this is a very relative/subjective way of description, it is essential for the understanding of the reader to give a more detailed explanation of which specific parameter define a ‘mega’-cluster. What were the criteria for defining specifically >800 molecules as a mega cluster? It might be better to just call them ‘clusters’ as this, in itself, is a general terminology to define the spatially confined accumulation of proteins. Indeed, mentioning the existence a broad distribution of oligomeric forms would describe better the results obtained and help a more accurate mechanistic understanding. At 120 min the mean cluster size is below 70 molecules, indicating that the large clusters above 800 molecules are minor oligomeric forms.

- Generally, the subpanels of the figures (mainly main figures) appear rather small and very compact. Especially the size and contrast of the representative microscopy images is not sufficient to observe the results. The images need to be improved in brightness and contrast (Fig. 1 E, J and M, Fig. S3 A, Fig. S6 A, and especially Fig. 2 C and H, Fig. 3 D, Fig. S7 B, and Fig. S9 B)

In addition, the authors need to make sure to check for signal saturation (Fig. 3 J, K, and M, Fig. S1 B, Fig. S8 A, B, and E)

Minor comments:

- The authors should include a heading for the introduction and provide a much broader overview of the physiological and therapeutic relevance of the cGAS-STING pathway as well a more detailed overview of its regulatory steps.

- The authors state that “such large STING clusters were also observed after stimulation with HT-DNA transfection (fig. S3)”. Please quantify the cluster size in order to support this conclusion. Additionally, it would be beneficial to show pathway activation by WB after HT-DNA transfection to demonstrate the correlation between cluster size and downstream

signaling as in Fig. 1 F to I.

- The loading control (α -tubulin) of western blots is not equal and should be substituted by another housekeeping gene or the authors should show Ponceau-S staining for comparison (especially Fig. 2 B and Fig. 3 A, B)

- The authors state that the 'decrease of colocalization of mEos4b-STING and TGN38 120 min after stimulation' suggests that STING reached recycling endosomes. Despite being a valid assumption to explain the disappearance of STING from the TGN, there are no data shown in this study to support this assumption. The authors should make clear that this is a hypothetical assumption based on previously published models of STING signaling.

- The authors also refer to "some clusters containing 'even' >800 molecules of STING". Unless there is a specific reason for the threshold of 800 molecules, the word 'even' is subjective and should be removed.

- In the discussion, the authors state that their "proved the STING clustering model [...] and further revealed the molecular driving forces regulating the clustering" Further they state that their results show "why STING can recruit and activate TBK1 specifically at the TGN, not the other compartments including the ER and the Golgi". The authors provide clear evidence of STING clustering at the TGN and recruitment of TBK1 to these clusters for activation, however, there is no evidence for what is the molecular driving force for this process. It would be beneficial to speculate further which process could provide the energy to drive this process as well as to discuss which parameters cause the clustering of STING specifically at the TGN (considering that small clusters are found at the CGN in early stages of STING activation).

- Please describe the function of di-4-ANEPPDHQ in more detail and specify to which 'environmental factor' it is sensitive to. Furthermore, explain better how this probe is used to determine general polarization and what specific information this values represents.

- Please explain the choice of a 100 nm radius for the analysis in Fig. 4 F given that the

single-molecule localization precision of mEos4b was determined as 20 ± 1 nm.

- The sentence “The treatment with OSW-1 partially suppressed [...] and abolished the colocalization of TBK1-mScarlet1 and mNeonGreen-iD4H.” should be rephrased. It is clear that the colocalization is abolished as there is no iD4H signal after OSW-1 treatment.

- The authors should include the quantification of supplementary video 1 in the figures. The GP value at TBK1 foci in the video seems higher (magenta color, GP value close to 1) than in the quantification. Please check this quantification or choose a movie that is more representative. Also, consider to include a Golgi marker to prove that the phenomenon of increased GP at TBK1 foci happens at Golgi membranes.

- For better transparency, it would be good to include individual data points (of individual cells or biological replicates) instead of/in addition to bar graphs (especially for Fig. 2 E, F, J, K, Fig. 3 F, G, Fig. 4 E, G)

- In the sentence “We also notes the stimulation-dependent increase of the signal of mNeonGreen-iD4H ...” please specify that you refer to ‘DMXAA-induced stimulation’ and clarify that ‘perinuclear region’ refers to Golgi (Rab6-positive) membranes.

- Please specify in the text which control is used in the CBC analysis of STING and CGN/TGN (Fig. 1K, N). Also, please indicate the number of replicates in the figure.

- The authors should make sure to define all abbreviations used in the main text (e.g. PALM, OSBP, ...)

- It would be helpful to clarify in the main text that the “First we performed data acquisition using purified mEos4b ...” is the calibration for the actual experiment that is described later in the text.

- The authors should specify in the main text by which means mEos4b-STING was reconstituted in STING KO MEFs.

- In the sentence “We stimulated cells with an ‘agonist’ DMXAA and obtained live-cell PALM images for up to 120 min ...” please make clear that it is a STING. Furthermore, clarify whether “for up to 120 min” refers to the total measurement time, time of treatment with DMXAA, or the acquisition time.
- The sentence “In the following experiments, we analyze the ‘initial’ event of STING clustering, i.e., the one that occurred at the TGN” should be rephrased as the ‘initial’ clustering of STING happens at the CGN as shown in Fig. 1 J to L.
- Make sure to unify the abbreviation for ‘figure’ (e.g. “Fig.” with capital “F”)
- In the sentence “We recently ‘hypothesized’ that STING palmitoylation, ...” the word ‘hypothesized’ should be exchanged to e.g. ‘demonstrated’ as it was shown in a previous publication.
- Please provide a reference for the statement “The disease is caused by heterozygous mutations [...] which is essential for retrograde transport from Golgi to ER.”
- The authors should rephrase the sentence “Besides the inhibitory effect on cultured cells, OSW-1 ...” and specify that the inhibitory effect refers to ‘the pathway activation in cells’.
- In the sentence “We also notes the stimulation-dependent increase of the signal of mNeonGreen-iD4H ...” please specify that you refer to ‘DMXAA-induced stimulation’ and clarify that ‘perinuclear region’ refers to Golgi (Rab6-positive) membranes.
- In the sentence “The puncta of TBK1-mScarletI, at which TBK1 is activated ... with mNeonGreen-iD4H.” please specify that this is upon DMXAA treatment.
- The sentence “The treatment with OSW-1 partially suppressed [...] and abolished the colocalization of TBK1-mScarletI and mNeonGreen-iD4H.” should be rephrased. It is clear that the colocalization is abolished as there is no iD4H signal after OSW-1 treatment.

- The authors should include the quantification of supplementary video 1 in the figures. The GP value at TBK1 foci in the video seems higher (magenta color, GP value close to 1) than in the quantification. Please check this quantification or choose a movie that is more representative. Also, consider to include a Golgi marker to prove that the phenomenon of increased GP at TBK1 foci happens at Golgi membranes.

- Please specify the abbreviations Toff, Ton, etc in Fig. 1B in the figure legend.

- In Fig. S 6 A, please specify the the microscopy images show mEos4b-STING and that the COP variants are the cellular context.

- Include a western blot in Fig. S10 to demonstrate the efficiency of CH25H knock-down.

Reviewer #3 (Remarks to the Author):

Activation of STING requires cholesterol-mediated mega-clustering at the TGN

In this manuscript, the authors utilize single molecule imaging to find that STING becomes “mega-clustered” at the TGN after activation. Further, the authors propose that this action is required for STING to act as a scaffold for TBK1, allowing for subsequent phosphorylation of TBK1, IRF3 and STING. The authors report that palmitoylation of STING and cholesterol is required for STING mega-clustering. While the ideas put forth are interesting, the claims made by the authors are not fully substantiated by their data and much has been published elsewhere.

1) Authors report that STING colocalized with the CGN after 10 mins, and the TGN after 30 mins by PALM (Figure 1 J, M). This reviewer is having difficulty distinguishing between time points with chosen images. Further, in Figure S1B, STING appears to colocalized with GM130 and TGN38 to the same extent over a time course up to 120 mins.

2) In Figure 2 A/B, the effects of 2-BP and H-151 have largely been shown by previously

published works by the authors and others (PMID: 27324217, PMID: 33615891)

3) Figure 2C and H are too dark to evaluate on a screen or paper.

4) The authors demonstrated the importance of STING C88/91S in their previous work
PMID: 27324217

5) For COPA deficient cells, some reports indicate that ER-to-golgi trafficking is also impaired since COPII cannot be recycled back to the ER. Please discuss.

6) In figure 3A, the author show that 25HC can dampen pTBK1, pSTING and pIRF3. The authors did not include the concentration of 25HC in their methods or how it was delivered to cell culture. Additionally, they did not validate that 25HC was able to influence total or synthesized pools of cholesterol in their model system. For example, how would imaging performed in Fig 3 J be altered with 25-HC?

7) How do CH25H KO MEFs/macrophage respond to STING ligands?

8) In 3B, authors use OSW-1 to block cholesterol transport from the ER to the golgi, which lowers sting-mediated pTBK/pIRF3. Does this result in build up of cholesterol at the ER? Is OSW toxic to cells or induce apoptosis? What concentrations are utilized?

9) The use of in vivo OSW is a bit concerning. What cells do the authors think they are examining from the spleen? I would suspect that these cells are mostly T and B cells with a small portion being innate immune cells. Perhaps a better tactic would be to look at peritoneal macrophage? Or sorting cells from the spleen to exclude adaptive immune cells. For example, OSW may just be killing a population of immune cells from the spleen.

10) Figure 3D is impossible to interpret because too dark

11) The cell free cholesterol system in Figure 3i is misleading and not relevant in this reviewer's opinion. Why not feed cholesterol back to live cells with MBCD? Presumably you would be able to use your Neon Green tracker in figure 3J-M to determine cholesterol uptake into cells.

12) Is the increase in NeonGreen from lipid droplet formation? Or free cholesterol? Can it distinguish between cholesterol and cholesteryl-esters?

13) Authors describe filipin staining in their methods, but I do not find it in their figures or in referenced in their text elsewhere.

14) I think that further explanation of some of the methods/reagents and what's already known about STING oligomerization might provide more context for this work.

Reviewer #1 (Remarks to the Author):

In this study, the Tomohiko Taguchi group identifies the activation of STING requires cholesterol at the TGN for aggregation. And they utilize single-molecule imaging to demonstrate that STING becomes mega-clustered at the trans-Golgi network. With this technology, the aggregation of STING (in this paper called mega-cluster of STING) was monitored with the treatment of a palmitoylation inhibitor. Moreover, the recruitment of TBK1 by STING aggregation was directly observed using this technology. The authors got the same conclusion from this technology as the previous studies. The novel information the authors get from this study is that cholesterol is another layer of regulation of STING clustering and activation. Overall, the authors provide a cutting-edge technology to characterize the activation of STING and some novel information on STING activation regulation. The issues are listed below.

1. Does the STING cluster specifically emerge in TGN? Given that the TGN marker, e.g. TGN38/46, also exist in scattered endosomes, the author should test whether the STING cluster also locates in endosomes (may test by EEA1) or other related organelle.

Thank you for your thoughtful comments. We recently reported that STING translocated sequentially from the ER to lysosomes through the Golgi and recycling endosomes (Kuchitsu et al., *Nat Cell Biol.*, 25, 453-466, 2023). STING reaches recycling endosomes positive with transferrin receptor (TfnR) and Rab11 after DMXAA stimulation for 3-6 h.

Thus, we monitored the STING cluster up to 6 h. The new data are included in the revised Figure 1i-k. These data indicated that the STING cluster lasted during its traffic through recycling endosomes.

To further confirm the presence of the STING cluster at recycling endosomes, we prepared MEF cells expressing mEOS4b-STING/Halo7-Rab11 or mEOS4b-STING/TfnR-Halo7 and performed dual-color super-resolution microscopic observations. We found that STING clusters started to co-localize with these endosomal proteins 2-6 h after DMXAA stimulation as shown in the revised Figure 3. Considering the original data that STING clusters co-localized with TGN38 (30-60 min after DMAA stimulation), these data indicated that the STING cluster emerged at the TGN, not at (recycling) endosomes.

2. In figure 4 J K M, the cholesterol level seems to be induced after STING stimulation, what is the possible reason? Is iD4H only probe for cholesterol detection? Other probes should be used to confirm this observation.

Thank you for the critical comments on the alteration of cholesterol levels. As far as we know, there are two cholesterol probes, a polyene macrolide antibiotic filipin and iD4H (and its derivatives). Filipin is commonly used but photo-bleached immediately. Therefore filipin is used to examine the subcellular distribution of cholesterol, but not to quantitate the amount of cholesterol. Filipin binds cholesterol in both leaflets of membranes. In contrast, iD4H, because it is expressed in the cytosol, binds to cholesterol in the cytosolic leaflet of membranes.

We originally quantitated the Pearson's correlation coefficient (PCC) between STING and iD4H, or between Rab6 and iD4H (Figure 3L, 3N, and S8E in the original manuscript). In the revised experiments, we quantitated the intensity of iD4H in the Rab6-positive area to address to the reviewer's suggestion. We found that the amount of iD4H in the Rab6-positive area increased significantly after stimulation (in the revised Figure 7i-k). Very excitingly, 2-BP (pan-palmitoylation inhibitor) and H-151 (STING palmitoylation inhibitor) suppressed the increase. Thus, we assume that palmitoyl groups on STING drives the cholesterol accumulation in the cytosolic leaflet of the TGN membranes.

We also used filipin to monitor the cholesterol distribution. As shown in the revised Figure S10b, DMXAA treatment did not increase the co-localization of filipin and Rab6. These results suggested that DMXAA did not increase the cholesterol levels at the TGN. OSW-1 (an inhibitor of cholesterol transport from the ER to the Golgi) reduced the co-localization (Figure S10b), confirming that OSW-1 indeed reduced the cholesterol levels in the TGN and validating the method of cholesterol imaging with filipin.

Please be noted that iD4H binds to cholesterol-containing liposomes sigmoidally with IC_{50} of 30% (Figure S11a and b in the revised manuscript). This binding character of iD4H to cholesterol may explain, in part, the discrepancy of the results with two cholesterol probes.

As mentioned, filipin does not discriminate the cholesterol in the cytosolic and luminal leaflets and does not quantitate the amount of cholesterol accurately. However, we believe that the combinatorial results with iD4H and filipin are decent enough to suggest the local accumulation of cholesterol triggered by STING palmitoylation.

In sum, we revised the manuscript as follows:

- (i) The original Figure 3J-N (in which iD4H was used to measure the PCC) was replaced with the new Figure 7i-k (in which iD4H was used to measure the amount of cholesterol in the Rab6-positive area).
- (ii) We included the new Figure S10b related to the new Figure 7.

3. In figure S1A, the expression level of STING in STING KO MEFs is inconsistent with p-TBK1 and p-STING.

Thank you for the comment. We performed WB several times and confirmed that mEos4b-STING, compared to endogenous STING, was less potent to activate TBK1 as shown in Figure S1a. We commented on this in the revised manuscript (page 7 line 10).

4. Please repeat this experiment in Fig 2A, 2B, 2G, 3B, 3I, S7A and S10B, total protein level of IRF3, TBK1, STING and tubulin are not in a similar level.

We repeated the experiments and the new results are now shown in Figure 4a, 4b, 4g, 6a, 7b, 7h, and S14b. The amount of IRF3 in CH25H-KD cells was found to be consistently less, compared to the amount of IRF3 in control siRNA-treated cells (Figure S14b).

5. The authors should reconsider the statement in page 7, line 9. Why discuss cGAS and cGAMP? There is no connection with the theme. If you want to exclude cGAS and cGAMP, cGAS KO cells should be included.

Thank you for the suggestion. As we would like to examine and exclude the contribution of cGAS and cGAMP to the STING clustering, we prepared cGAS KO cells stably expressing mEos4b-STING. We then performed PALM observation in these cells with empty vector (mock), wild-type (WT) α -COP, or the D243G mutant. We found that the expression of the D243G mutant induced large mEos4b-STING clusters, while mock and WT did not (Figure S7).

6. In fig 3A and 3B, p-STING and p-IRF3 seemed to be inhibited obviously, but p-TBK1 was not affected dramatically, please explain.

Thank you for the critical comments. As pointed out, p-TBK1 was less affected in the conditions where the Golgi cholesterol was reduced (Figure 7a and b). In contrast, p-TBK1 was dramatically reduced in the conditions where palmitoylation of STING was inhibited (Figure 4a and b). Thus, we assume that orientation of individual STING molecules in the small STING cluster may differ in the two conditions. In cholesterol-depleted conditions, even a small STING cluster may accommodate two TBK1 dimers simultaneously, thus still facilitating autophosphorylation of TBK1. In contrast, a small cluster that palmitoylation-deficient STING forms may not accommodate two TBK1 dimers. We described these possibilities in the revised manuscript in the Discussion section (page 16 line 1).

7. Page 8, line 10, please add a control without TBK1. In fig 3I, cholesterol was essential for priming STING for its phosphorylation, but how can you examine whether STING is primed for the phosphorylation by TBK1.

Thank you for the critical suggestion. A control without recombinant TBK1 was added (lane 2 in Figure 7h), showing the level of TBK1-independent STING phosphorylation in the assay. With this control, we believe that a set of data in Figure 7h suggests that cholesterol is essential for priming STING for the phosphorylation by TBK1.

In related to this, please be noted that another reviewer asked us to do cholesterol add-back experiment in live cells. As shown in Figure S9d, cholesterol add-back to cells restored pTBK1/pSTING/pIRF3, suggesting that the inhibitory effect of 25-HC and OSW-1 was through the reduced cholesterol levels.

8. Page 9, line 6, please explain why you test the signal of mNeonGreen-iD4H at the perinuclear region. Does cholesterol traffic with STING to endosome?

Thank you for the comment. As the reviewer1 suggested in the critique 2, we performed the new experiments in which the intensity of iD4H on Rab6-positive regions was quantitated. We examined the Rab6-positive area, because this site is where STING forms clusters.

Regarding the latter question, we examined the signal of iD4H and TfnR. There appeared some overlaps between STING, TfnR, and iD4H after stimulation. The iD4H signal was very low before stimulation (as shown in Figure 7i and j) and we did not see the co-localization between iD4H and TfnR. These results suggested that a fraction of cholesterol moved with STING to recycling endosomes. However, to conclude this, more experiments are required and we would like to reserve this issue for the future studies.

Localization of STING, iD4H and TfnR after DMXAA stimulation.

Sting^{-/-} MEFs expressing mRuby3-STING, mNeonGreen-iD4H and TfnR-Halo were stimulated with DMXAA (25 $\mu\text{g mL}^{-1}$) for 2 h. Cells were fixed. Scale Bar, 10 μm .

9. Fig S1B 和 S8A, the mEos4b-STING clusters colocalized with GM130 and TGN38 is different in Fig S1B. It is not consistent.

Thank you for the critical comments. Our purpose of these experiments was just to show that mEos4b-STING was functional in terms of translocation (Figure S1B in the original manuscript), or to show that STING translocated to the Golgi in cholesterol-depleted conditions (Figure S8 in the original manuscript). So we used Zeiss airy-scan microscopy for these experiments, the spatial (XY) resolution of which is about 120 - 250 nm. With this resolution, we cannot discriminate CGN (GM130) and TGN (TGN38). On the other hand, the spatial (XY) resolution of the PALM image is about 20 - 30 nm. With this resolution, as shown in the revised Figure 2, CGN (GM130) and TGN (TGN38) can be segregated.

With this critical comment, we realized that Figure S1B and S8A with airy-scan microscopy were confusing in the context of the present study. We thus revised these Figures as follows:

- (i) Figure S1B was deleted.
- (ii) The staining of GM130 was deleted from Figure S8A and the results were shown in the revised Figure S9a and b.

10. OSBP should use the full name when it is first mentioned.

Thank you for the comment. OSBP (oxysterol-binding protein) was spelled out when it is first mentioned.

Reviewer #2 (Remarks to the Author):

In the manuscript entitled "Activation of STING requires cholesterol-mediated mega-clustering at the TGN", Kemmoku et. al. investigate the regulation of STING activation – a crucial step in the cGAS-STING signaling pathway which is a key mediator of inflammation in the context of infection, cellular stress and tissue damage. The authors employ quantitative photoactivated localization microscopy (PALM) of reconstituted mEos4b-STING in MEF cells to measure cluster formation of STING. Co-staining with cis- and trans-Golgi markers revealed a time-dependent appearance of large STING clusters at the trans-Golgi network after treatment of the cells with the STING agonist DMXAA. The authors demonstrate a correlation between the average size of STING clusters and the degree of downstream pathway activation. This is in agreement with their observation that STING clusters facilitate the recruitment of the downstream signaling molecule TBK1 and presumably its activation at STING clusters. Using palmitoylation inhibitors as well as a palmitoylation-deficient mutant of STING, the authors found a dependency between palmitoylation of STING and cluster formation. Furthermore, the authors show that STING clusters formation and the degree of downstream pathway activation is modulated by the presence of cholesterol at the Golgi membranes and that STING clusters localize to areas of increased membrane order. Based on these findings the authors propose a model of palmitoylation and cholesterol-dependent STING clustering in membrane microdomains at the trans-Golgi network that has a direct correlation with the degree of activation of the signaling pathway. The study is very interesting for the field in two ways, because it succeeds at linking oligomerization measured directly in cells with function, and because it advances our understanding of the molecular mechanisms in the STING pathway.

Major comments:

- For quantification of STING clusters, the authors apply a previously described kinetic model for the quantification of PALM data. The accuracy of the kinetic model relies on the accuracy of the calibration and the correct determination of the photophysical properties of the used fluorophore (Nblink, Ton, Toff, etc). Here, the authors used PVA-embedded mEos4b monomers on glass for calibration. The physical and chemical environment of a fluorophore or fluorescent protein might greatly influence its photophysical properties (transition rates from fluorescent to dark state, blinking rates, photobleaching, etc). It is therefore of essential importance to account for differences of these properties in the calibration sample compared to the actual measurement sample (i.e. mEos4b in PVA on glass for calibration vs mEos4b tagged to STING in cells).

It is important to discuss the possible influence of PVA embedding on the fluorescent properties in the text/methods as a limitation or to compare the measured calibration values to the ones obtained from a calibration with different sample preparation (e.g. biotinylated mEos4b on streptavidin-coated glass coverslips as in the original publication (Lee et. al., PNAS, 2012)).

Thank you very much for your considerate remarks. Thedie et al. showed that the photoswitching behaviors of mEOS4b, mEOS3.2, and mEOS2 in PVA are similar to those in bacterial cells although they could not examine if one of the photophysical parameters, the recovery rate of short-lived dark state, D_{short} in PVA is similar to that in cellular environment (*J. Phys. Chem. Lett.*, 8, 4424-4425, 2017). Therefore, they could not exclude the possibility that PVA restricts oxygen accessibility and exerts redox effects. We mentioned this and cited this reference in the revised version.

Nevertheless, as shown in the revised version (Figure 1f and g), we have succeeded in approximating the mean quantity, estimated to be approximately 50, of mEOS4b-cavin1 molecules in each discrete caveolae structure. This finding is consistent with the previously reported value that was estimated by FCS (Gambin et al., *eLife*, 3, e01434, 2014). Furthermore, analysis by model-based correction (MBC) (Jensen et al., *Nat. Methods*, 19, 594-602, 2022), which obviates the necessity for direct mEOS4b observation in PVA, showed the average number of STING in a cluster, which is similar to that estimated by using the photophysical parameters characterizing mEOS4b blinking obtained in PVA (Figure S5 in the original version and Figure S4 in the revised version). These results collectively affirm the successful and largely accurate estimation of the abundance of mEOS4b-STING complexes in clusters through the utilization of photophysical parameters characterizing mEOS4b in PVA.

- Additionally, for accurate calibration the authors should show only monomeric mEos4b is measured. It would be helpful to show a FWHM distribution or brightness distribution of the measured molecules to demonstrate that mEos4b was monomeric.

Thank you very much for your helpful suggestion. The distribution of the fluorescence intensity of mEOS4b in PVA, could be only fitted with a single lognormal function, based on both Akaike and Bayesian information criteria (Tsunoyama et al., *Nat. Chem. Biol.* 14, 797-506, 2014). Therefore, we concluded that only the mEOS4b monomer was observed. We included a new Figure 1c of the distribution of fluorescence intensity of mEOS4b.

- However, the question whether oligomerization into clusters of hundreds of molecules changes the photophysical properties of mEos4b compared to the monomeric form should be addressed as it would also have a strong impact on the quantification of the oligomeric state. More adequate calibration references using oligomers of mEos4b of known stoichiometry that are closer to the oligomer size investigated here would be desirable.

Thank you very much for your thoughtful comments. We agree with the reviewer's discerning comment as it promises to substantively augment the methodological robustness. PC3 cells express caveolin but remain devoid of cavin1, thereby resulting in the absence of caveolae. Meanwhile, previous reports showed that the transfection of PC3 cells with cavin1 cDNA induced the formation of caveolae (Hill et al., *Cell*, 132, 113-124, 2008). Consequently, we performed PALM observations of mEOS4b-cavin1 in PC3 cells because the labeling efficiency of cavin1 with mEOS4b is 100%. Both segmentation methods, including Kernel Density Estimation (KDE) and SR-Tesseler, revealed that the numbers of mEOS4b-cavin1 molecules in individual caveolae were estimated to be approximately 50 (52 ± 2 and 49 ± 2 , respectively). These values are consistent with the previously reported one (50 ± 5) measured by FCS (Gambin et al., *eLife*, 3, e01434, 2014). These results are shown in Figure 1f and g in the revised version.

- The size of STING clusters at the TGN might be partially determined by the abundance of Golgi membrane and the STING expression levels (especially for late time points after activation). It would be interesting to check for this correlation by normalizing the size of STING clusters to a TGN signal and the absolute cellular abundance of STING.

Thank you for the thoughtful comment. We agree with the reviewer and counted the number of localizations of mEOS4b-STING and TGN-Halo7-SF650B after DMXAA stimulation for 60 min. Subsequently, we undertook an assessment of the correlation between the amounts of mEOS4b-STING or TGN-Halo7-SF650B and CBC score. Remarkably, Spearman's correlation coefficients for all instances fell within the range of -0.3 and 0.3, denoting an exceedingly feeble correlation. In the revised version, we have incorporated the histograms in Figure S6 in the revised version.

- The dependence of STING signaling on the presence of cholesterol at the Golgi is

demonstrated by a reduction of TBK1-phosphorylation in cells, a decrease in downstream mRNA expression in vivo and a reduction of the number of STING clusters upon OSBP inhibition (Fig. 3 A to G). It is further supported by an in vitro cholesterol reconstitution assay (Fig. 3 H, I). It is highly important to control the effect of cholesterol depletion at the Golgi (after OSW-1 or 25-HC treatment) on overall Golgi size, membrane abundance and membrane integrity at the Golgi as well as its overall 'functionality'.

Thank you for the comments. As mentioned in the following one-by-one responses, we addressed to these critiques individually.

- Despite the reduction in the pathway activation and STING cluster number and size under OSW 1 treatment conditions (Fig. 3 A, B), in Fig. 3 J, K and M it is evident that both, STING and TBK1 still cluster at perinuclear (Golgi) regions under these conditions, indicating that cholesterol might influence but is not required for their clustering. The authors should discuss this observation in more detail and avoid describing the role of cholesterol as 'essential'. Furthermore, it is important to control that the reduction in clustering is not consequence of overall reduced Golgi membrane abundance after cholesterol depletion as mentioned above.

We agreed with the critique and revised the texts as cholesterol functions as a facilitating factor for STING clustering.

To address the latter question, we measured the Rab6-positive area with/without cholesterol depletion. As shown in the revised Figure S11f, the Rab6-positive area was not grossly affected with OSW-1 treatment.

- The authors show that the staining with mNeonGreen-iD4H can be used to detect abundances of cholesterol of higher than 20 % (Fig. S8 D). Treatment with DMXAA for 60 min leads the detection of mNeonGreen-iD4H at Golgi regions (Rab6-positive membranes, Fig. 3 M) indicating that the induction of STING signaling using DMXAA triggers the enrichment of cholesterol at the Golgi. The authors should test or discuss if this is caused by enhanced cholesterol transport to Golgi membranes (e.g. through the secretory pathway) or reorganization of Golgi cholesterol into enriched domains.

Thank you for the critical suggestions. We originally quantitated the Pearson's correlation

coefficient (PCC) between STING and iD4H, or between Rab6 and iD4H (Figure 3L, 3N, and S8E in the original manuscript). In the revised experiments, we quantitated the intensity of iD4H in the Rab6-positive area. We found that the amount of iD4H in the Rab6-positive area increased significantly after stimulation (in the revised Figure 7i-k). Very excitingly, 2-BP (pan-palmitoylation inhibitor) and H-151 (STING palmitoylation inhibitor) suppressed the increase. Thus, we assume that palmitoyl groups on STING drives the cholesterol accumulation in the cytosolic leaflet of the TGN membranes.

- The authors test for colocalization of mRuby3-STING, TBK1-mScarlet and Halo-Rab6 with mNeonGreen-iD4H (Fig. 3 J to N and Fig. S 8 E). In untreated samples (DMXAA 0 min) as well as after inhibition of cholesterol transport to Golgi membranes (OSW-1 treatment), hardly any mNeonGreen-iD4H signal is detectable. Without decent signal intensity, it is not possible to measure colocalization or to calculate trustable Pearson's correlation coefficients. Furthermore, the mNeonGreen-iD4H signal in the representative images of DMXAA-treated samples seems saturated which impairs the accuracy of the analysis as well. The authors should improve this analysis in order to make trustable conclusions or should quantify the signal intensity of mNeonGreen-iD4H as an indicator for the cholesterol abundance instead of colocalization analysis.

Thank you for the critical suggestion. As mentioned above, in the revised experiments, we quantitated the intensity of iD4H in the Rab6-positive area (in the revised Figure 7i-k).

- Based on the influence of (i) the presence of cholesterol on Golgi membranes (Fig. 3), (ii) the palmitoylation of STING (Fig. 2) and (iii) the increase of lipid order/membrane rigidity on STING clustering and downstream pathway activation (Fig. S7), the authors propose in their model the requirement of the formation of lipid microdomains ("rafts") for STING activation. While the phenomenon of "lipid rafts" is debated for decades as a hub for the enrichment and signaling of proteins, the author should be careful with their statement on the essentiality of lipid rafts for STING activation. In order to make this conclusion, it is essential to demonstrate that the formation of these lipid microdomains is a causative regulatory step for the pathway execution and to rule out that their formation is a consequence of it (i.e., the possibility that STING palmitoylation and its putative intrinsic property for clustering or oligomerization upon activation causing the reorganization of the membrane into a specialized micro-environment.

Thank you for the comments. As suggested, we interfered with the palmitoylation step of STING and found that it was the key factor to enrich iD4H on the TGN membranes (in the revised Figure 7i-k). These results suggested that palmitoyl groups on STING drives the reorganization of Golgi cholesterol into enriched domains.

- In Fig. 4 A to D, the authors show the transient localization of TBK1 monomers to STING clusters upon DMXAA treatment with a colocalization duration of ~150 ms. How do the authors explain this transient localization and dissociation compared to the formation of large (stable) clusters of TBK1 as shown in Fig. 3 K?

In Fig. 4A-D of the original version, in order to observe single molecules of TBK1-Halo7-SF650T, the fluorescence labeling efficiency of TBK1-Halo7 should be very low and the concentration of TBK1-Halo7-SF650T in the cytosol should be less than 1~2 nM. Under this condition, we found that the individual TBK1 molecules dynamically bound to STING clusters for short periods upon DMXAA stimulation. On the other hand, since the number of TBK1-mScarlet1 molecules in a cell in Fig. 3K of the original version was much higher than that of TBK1-Halo7-SF650T molecules in Fig. 4A-D, large clusters of TBK1 could be observed in Fig. 3K. Even if large clusters of TBK1 are formed, the residency time of individual TBK1 molecules on STING clusters was very short. Such structures have often been found in the cell plasma membrane. For example, immunofluorescence staining shows cluster formation of integrin β 1, meanwhile, single-molecule observation revealed that integrin β 1 molecules were transiently confined in focal adhesion for short periods (Tsunoyama et al., *Nat. Chem. Biol.*, 14, 497-506, 2018). In another example, epi-fluorescence or confocal microscope has shown that Grb2 clusters are colocalized with EGFRs upon stimulation. However, single-molecules of Grb2 were only transiently (~120 ms) recruited to EGFR (Morimatsu et al., *PNAS*, 104, 18013-18018, 2007). Therefore, the large clusters shown in Fig. 3K (Fig. S11c in the revised version) are formed by transient recruitment of TBK1 to STING clusters.

- The authors demonstrate the increase of the number of STING clusters caused by disease-causative COPA variants in absence of the STING agonist DMXAA. This result indicates that COP-I-mediated retrograde transport might be involved in the reduction of STING clusters by removing STING from the CGN but doesn't provide evidence for the role of STING clustering on the outcome of the signaling pathway. The authors should provide data on the activation of the signaling pathway in presents of these COPA variants and demonstrate how

STING is activated in absence of the STING agonist DMXAA or cGAMP under these conditions.

Thank you for the critical comments. In the revised manuscript, we performed the WB analysis and showed that the STING pathway was activated in the presence of the disease-causative COPA variants (Figure 5d in the revised manuscript). Please be noted that the activation of the downstream genes was reported in our previous paper (Mukai et al., *Nat Commun* 12, 61, 2021).

- The authors measured the number of STING molecules in a radius on 100 nm around TBK1-Halo spots and compared it to those around 'randomized' TBK1 spots in silico. This methodology needs a much broader introduction and explanation. How are the TBK1 spots randomized? What are the parameters and assumptions? How is the localization of STING measured relative to the simulated TBK1 spots?

We agree with the reviewer's comments. To explain the methodology more comprehensively, we described more details of the methods in the Supplementary Method section of "*Simultaneous live-cell PALM of mEOS4b-STING, to visualize STING clusters and single-molecule imaging of TBK1-Halo7, to track their recruitment to STING clusters, and estimation of the residency lifetimes of TBK1-Halo7 inside the STING cluster*". Here, we described that the TBK1 spots are randomized by shifting the trajectories of TBK1-Halo7-SF650T in random directions by random distances. Furthermore, the superimposition of two images taken by two different cameras was performed as we previously reported (Suzuki et al., *J. Cell Biol.* 177, 717-730, 2007). We can determine the localization of STING relative to the stimulated TBK1 spots and count the number of mEOS4b-STING around the single molecules of recruited TBK1-Halo7-SF650T.

- The authors refer to "STING 'mega'-clusters" several times throughout the manuscript. As this is a very relative/subjective way of description, it is essential for the understanding of the reader to give a more detailed explanation of which specific parameter define a 'mega'-cluster. What were the criteria for defining specifically >800 molecules as a mega cluster? It might be better to just call them 'clusters' as this, in itself, is a general terminology to define the spatially confined accumulation of proteins. Indeed, mentioning the existence a broad distribution of oligomeric forms would describe better the results obtained and help a more

accurate mechanistic understanding. At 120 min the mean cluster size is below 70 molecules, indicating that the large clusters above 800 molecules are minor oligomeric forms.

In the original manuscript, we focused on the cluster containing >800 molecules because it emerged 10 min after stimulation (Figure 1F in the original manuscript). However, as the reviewer pointed out, we agreed that this was a very subjective way of description. We thus simply call them "clusters" in the revised manuscript.

- Generally, the subpanels of the figures (mainly main figures) appear rather small and very compact. Especially the size and contrast of the representative microscopy images is not sufficient to observe the results. The images need to be improved in brightness and contrast (Fig. 1 E, J and M, Fig. S3 A, Fig. S6 A, and especially Fig. 2 C and H, Fig. 3 D, Fig. S7 B, and Fig. S9 B). In addition, the authors need to make sure to check for signal saturation (Fig. 3 J, K, and M, Fig. S1 B, Fig. S8 A, B, and E)

Thank you very much for checking all of the images carefully. We improved all the images in brightness, contrast, and saturation. Please be noted that images in Figure 4c, 4h, 5a (mock and WT), 6b (D-Cer-C6), 7c, S7a (mock and WT) in the revised manuscript are dim because of the reduced STING clustering. The size of Figure 4c, 4h, 6b, 7c, S13b in the revised manuscript was expanded.

Minor comments:

- The authors should include a heading for the introduction and provide a much broader overview of the physiological and therapeutic relevance of the cGAS-STING pathway as well a more detailed overview of its regulatory steps.

Thank you for the comments. We revised the introduction section accordingly (page 4).

- The authors state that "such large STING clusters were also observed after stimulation with HT-DNA transfection (fig. S3)". Please quantify the cluster size in order to support this conclusion. Additionally, it would be beneficial to show pathway activation by WB after HT-DNA transfection to demonstrate the correlation between cluster size and downstream signaling as in Fig. 1 F to I.

The results were shown in the original manuscript in Figure S3B. Because of the reformatting of the whole manuscript, the results are now shown in Figure S5 in the revised manuscript. STING activation by HT-DNA transfection was shown by the new Figure S5a.

- The loading control (α -tubulin) of western blots is not equal and should be substituted by another housekeeping gene or the authors should show Ponceau-S staining for comparison (especially Fig. 2 B and Fig. 3 A, B)

Thank you for the comments. All of the WB were carefully examined and replaced with new blots when necessary. The data in Figure 2B, 3A, and 3B in the original manuscript are now shown in Figure 4b, 7a, and 7b in the revised manuscript.

- The authors state that the 'decrease of colocalization of mEos4b-STING and TGN38 120 min after stimulation' suggests that STING reached recycling endosomes. Despite being a valid assumption to explain the disappearance of STING from the TGN, there are no data shown in this study to support this assumption. The authors should make clear that this is a hypothetical assumption based on previously published models of STING signaling.

Thank you for critical suggestions. To confirm the STING translocation from the TGN to recycling endosomes 2 h after stimulation, we prepared MEF cells expressing mEOS4b-STING/TfnR-Halo7 or mEOS4b-STING/Halo7-Rab11, and performed dual-color super-resolution microscopic observations. As shown in the new Figure 3c, the co-localization of STING with TfnR increased during 60 - 120 min after stimulation, which contrasted with the decreased co-localization of STING with TGN38 during that time (Figure 2d). These results showed that STING translocated from the TGN to recycling endosomes during 60 - 120 min after stimulation.

- The authors also refer to "some clusters containing 'even' >800 molecules of STING". Unless there is a specific reason for the threshold of 800 molecules, the word 'even' is subjective and should be removed.

Thank you for the suggestion. We deleted "even" from the corresponding sentence.

- In the discussion, the authors state that their “proved the STING clustering model [...] and further revealed the molecular driving forces regulating the clustering” Further they state that their results show “why STING can recruit and activate TBK1 specifically at the TGN, not the other compartments including the ER and the Golgi”. The authors provide clear evidence of STING clustering at the TGN and recruitment of TBK1 to these clusters for activation, however, there is no evidence for what is the molecular driving force for this process. It would be beneficial to speculate further which process could provide the energy to drive this process as well as to discuss which parameters cause the clustering of STING specifically at the TGN (considering that small clusters are found at the CGN in early stages of STING activation).

In the present study, we provided the evidence that STING palmitoylation, sphingomyelin, and cholesterol facilitate the STING clustering (Figure 4, 6, and 7 in the revised manuscript). Given that cholesterol and sphingomyelin are enriched in the TGN by direct transfer of cholesterol and ceramide (the substrate for sphingomyelin synthase) from the ER with the aid of OSBP and CERT (Nat. Rev. Mol. Cell Biol. 9, 273, 2008), we assume that the presence of these lipids specifically in the TGN accounts for the STING clustering there, not in the CGN.

We assume that clustering of STING is beneficial for TBK1 recruitment. Once TBK1 is detached from STING, there is other (TBK1-free) STING molecules around (if STING clusters), which increases the encounter probability of TBK1 and STING. Clustering of STING may also be beneficial for TBK1 activation, because two TBK1 dimers (TBK1 exists as a dimer) need to be placed close each other for autophosphorylation. STING cluster may act as a suitable platform to accommodate multiple TBK1 dimers, facilitating TBK1 activation (autophosphorylation).

- Please describe the function of di-4-ANEPPDHQ in more detail and specify to which ‘environmental factor’ it is sensitive to. Furthermore, explain better how this probe is used to determine general polarization and what specific information this values represents.

Di-4-ANEPPDHQ is a fluorescent probe that exhibits distinct changes in its emission spectrum in response to variations in lipid environment (Jin et al., *Biophys. J.* 90, 2563-2575, 2006). Notably, it is highly sensitive to the state of lipid packing within cellular membranes

(Owen et al., *Nat. Protoc.* 7, 24-35, 2011). An increase in the concentration of cholesterol in biomembranes enhances lipid packing, thus affecting the emission spectrum of Di-4-ANEPPDHQ.

Routinely, by the following equation, a generalized polarization (GP) value is determined.

$$GP = \frac{\text{Intensity}(500-580) - \text{Intensity}(620-750)}{\text{Intensity}(500-580) + \text{Intensity}(620-750)}$$

In the case where [Intensity (500-580)] >> [Intensity (620-750)], GP is close to 1.

In the case where [Intensity (500-580)] << [Intensity (620-750)], GP is close to -1.

As lipid packing increases, the fluorescence spectrum shifts toward shorter wavelengths, and GP values are accordingly higher. Conversely, GP values are lower in membranes with less lipid packing.

- Please explain the choice of a 100 nm radius for the analysis in Fig. 4 F given that the single-molecule localization precision of mEos4b was determined as 20 ± 1 nm.

As shown in Fig. 4F of the original version, we measured the numbers of STING molecules around TBK1-Halo7-SF650T molecules and those around randomized spots, and the former minus the latter was plotted in the histogram. Then, the average numbers in the consecutive positive value regions were estimated as shown in Fig. 4G (Fig. 8g in the revised version). However, if we employ a smaller radius such as 50 nm as an area to detect mEOS4b-STING numbers, the number of mEOS4b-STING becomes small, and the bars in the histogram fluctuate very much. We cannot perform the same analysis because negative values often appear. We described these on pages 33-34 in the Materials and Methods of the revised version.

- The sentence "The treatment with OSW-1 partially suppressed [...] and abolished the colocalization of TBK1-mScarlet1 and mNeonGreen-iD4H." should be rephrased. It is clear that the colocalization is abolished as there is no iD4H signal after OSW-1 treatment.

Thank you for the critical comments. We agree that the colocalization analysis is meaningless

as there is no iD4H signal after OSW-1 treatment. We revised the corresponding texts accordingly.

- The authors should include the quantification of supplementary video 1 in the figures. The GP value at TBK1 foci in the video seems higher (magenta color, GP value close to 1) than in the quantification. Please check this quantification or choose a movie that is more representative. Also, consider to include a Golgi marker to prove that the phenomenon of increased GP at TBK1 foci happens at Golgi membranes.

Thank you for the critical comments. In the original movie, the way of pseudo-coloring of the GP value in magenta did not have a good dynamic range. Thus, we revised the way of pseudo-coloring in a different way (blue to red) in the new Supplementary Movie S1. We also put the GP value in the TBK1-positive area (the middle panel in the Movie).

Regarding an inclusion of a Golgi marker, four colors (one color for the Golgi, one color for TBK1, two colors for GP) should be captured at the same time. However, our microscope can capture three colors at most, preventing us from performing the experiment.

- For better transparency, it would be good to include individual data points (of individual cells or biological replicates) instead of/in addition to bar graphs (especially for Fig. 2 E, F, J, K, Fig. 3 F, G, Fig. 4 E, G)

We thank the reviewer for this useful comment. We agree to include data points of individual cells in addition to the bar graphs (Figure 2E, 2J, and 3F in the original manuscript), and showed them in Figure 4e, 4j, and 7e in the revised manuscript. On the other hand, Figure 2F, 2K, and 3G in the original manuscript show the number of STING in larger clusters (>800)/ μm^2 . However, only small numbers of such large clusters could be observed in one cell, and the numbers varied widely from cell to cell. Therefore, we cannot show the data points of individual cells in Figure 2F, 2K, and 3G.

Figure 4E in the original manuscript shows the lifetimes of colocalization duration of STING clusters with TBK1. The colocalization lifetimes were estimated by curve fitting of the histograms with a single exponential decay function as shown in Figure 4D, and SEM in Figure 4E means the fitting error. If data in individual cells are fitted with a single exponential

decay function, the colocalization lifetimes vary widely from cell to cell due to small numbers of data points. Furthermore, it is usual to show the colocalization lifetimes estimated by fitting the data obtained in many cells (For example, Suzuki et al., *Nat. Chem. Biol.*, 8, 774-783, 2012; Komura et al., *Nat Chem Biol.*, 12, 402-410, 2016; Morise et al., *Nat. Commun.*, 10, 5245, 2019). Therefore, we did not show the data point of individual cells. The legend of the vertical axis of Figure 4E (mean \pm SEM) of the original version was misleading, so it was changed to "Colocalization lifetime (ms)" in the revised version (Figure 8e).

Figure 4G in the original manuscript shows an average number of STING molecules around TBK1 (100 nm radius) in the consecutive positive value regions in Figure 4F. If we plot the data obtained in individual cells, the number of mEOS4b-STING becomes small, and the bars in the histogram fluctuate very much, and we cannot perform the same analysis because negative values often appear. Therefore, we did not show the data point of individual cells in Figure 8g in the revised version.

- In the sentence "We also notes the stimulation-dependent increase of the signal of mNeonGreen-iD4H ..." please specify that you refer to 'DMXAA-induced stimulation' and clarify that 'perinuclear region' refers to Golgi (Rab6-positive) membranes.

Thank you for the comment. We revised the texts accordingly.

- Please specify in the text which control is used in the CBC analysis of STING and CGN/TGN (Fig. 1K, N). Also, please indicate the number of replicates in the figure.

We already specified which control is used in the analysis in the Figure 1K caption, "CBC scores relative to control (180° rotated image) are shown in red."

We added the number of replicates in the Figure caption.

Figure 1K (Figure 2b in the revised version): 10min n=10, 30 min n=12

Figure 1N (Figure 2e in the revised version): 10 min n=12, 30 min n=11, 60 min n=13, 120 min 2=12

Furthermore, we performed dual-color super-resolution observation of mEOS4b-STING and mTfnR or Rab11, and showed the results in the revised version. We indicate the number of

replicates in the Figure caption:

Figure 3b and e in the revised version: n=10 for all incubation periods

- The authors should make sure to define all abbreviations used in the main text (e.g. PALM, OSBP, ...)

We revised the texts accordingly.

- It would be helpful to clarify in the main text that the “First we performed data acquisition using purified mEos4b ...” is the calibration for the actual experiment that is described later in the text.

Thank you for this useful comment. We added the sentence “This is the calibration for the actual experiment that is described later.” in page 6 as the reviewer suggested.

- The authors should specify in the main text by which means mEos4b-STING was reconstituted in STING KO MEFs.

Thank you for the comment. We revised the text as follows (page 7 line 8):

"mEos4b-STING was stably expressed with virus transduction in STING KO MEFs"

- In the sentence “We stimulated cells with an ‘an agonist’ DMXAA and obtained live-cell PALM images for up to 120 min ...” please make clear that it is a STING. Furthermore, clarify whether “for up to 120 min” refers to the total measurement time, time of treatment with DMXAA, or the acquisition time.

Thank you for the comment. We revised the text as follows (page 7 line 11).

"We stimulated cells with a STING-agonist DMXAA for the indicated time and then obtained live-cell PALM images (Figure 1h-k).

- The sentence “In the following experiments, we analyze the ‘initial’ event of STING

clustering, i.e., the one that occurred at the TGN” should be rephrased as the ‘initial’ clustering of STING happens at the CGN as shown in Fig. 1 J to L.

Thank you for the comment. In the revised manuscript, in response to the reviewers' critiques, we analyzed the STING cluster up to 360 min after STING stimulation (Figure 1i-k). We carefully revised the whole texts that correspond to Figure 1i-k, 2, and 3 accordingly.

- Make sure to unify the abbreviation for ‘figure’ (e.g. “Fig.” with capital “F”)

We corrected them.

- In the sentence “We recently ‘hypothesized’ that STING palmitylation, ...” the word ‘hypothesized’ should be exchanged to e.g. ‘demonstrated’ as it was shown in a previous publication.

Thank you for the suggestion. We corrected it.

- Please provide a reference for the statement “The disease is caused by heterozygous mutations [...] which is essential for retrograde transport from Golgi to ER.”

Thank you for the suggestion. We included the reference (Watkin LB, et al., *Nat Genet*, 47, 654-660, 2015).

- The authors should rephrase the sentence “Besides the inhibitory effect on cultured cells, OSW-1 ...” and specify that the inhibitory effect refers to ‘the pathway activation in cells’.

Thank you for the comment. In response to the reviewer3's critique, we removed the results of OSW-1 administration to mice.

- In the sentence “We also notes the stimulation-dependent increase of the signal of mNeonGreen-iD4H ...” please specify that you refer to ‘DMXAA-induced stimulation’ and clarify that ‘perinuclear region’ refers to Golgi (Rab6-positive) membranes.

Please see the above. This is already asked.

- In the sentence “The puncta of TBK1-mScarletl, at which TBK1 is activated ... with mNeonGreen-iD4H.” please specify that this is upon DMXAA treatment.

We revised the texts accordingly.

- The sentence “The treatment with OSW-1 partially suppressed [...] and abolished the colocalization of TBK1-mScarletl and mNeonGreen-iD4H.” should be rephrased. It is clear that the colocalization is abolished as there is no iD4H signal after OSW-1 treatment.

Please see the above. This is already asked.

- The authors should include the quantification of supplementary video 1 in the figures. The GP value at TBK1 foci in the video seems higher (magenta color, GP value close to 1) than in the quantification. Please check this quantification or choose a movie that is more representative. Also, consider to include a Golgi marker to prove that the phenomenon of increased GP at TBK1 foci happens at Golgi membranes.

Please see the above. This is already asked.

- Please specify the abbreviations Toff, Ton, etc in Fig. 1B in the figure legend.

We specified the abbreviations in the Figure 1b caption.

Ton: fluorescence-on time before the molecule goes into the dark state or it photobleaches

Toff: fluorescence-off time

- In Fig. S 6 A, please specify the the microscopy images show mEos4b-STING and that the COP variants are the cellular context.

Thank you for the comments. We added "PALM images of mEOS4b-STING" on the top of

the images (Figure 5 in the revised manuscript).

In the original manuscript, we did not show the PALM image and data for mEOS4b-STING in the cell expressing wild-type α -COP. In the revised manuscript, we added the image and data in Figure 5.

- Include a western blot in Fig. S10 to demonstrate the efficiency of CH25H knock-down.

Thank you for the comment. We included the WB with anti-CH25H antibody (Figure S14b in the revised manuscript). The efficiency was estimated to be about 50%.

Reviewer #3 (Remarks to the Author):

Activation of STING requires cholesterol-mediated mega-clustering at the TGN

In this manuscript, the authors utilize single molecule imaging to find that STING becomes “mega-clustered” at the TGN after activation. Further, the authors propose that this action is required for STING to act as a scaffold for TBK1, allowing for subsequent phosphorylation of TBK1, IRF3 and STING. The authors report that palmitoylation of STING and cholesterol is required for STING mega-clustering. While the ideas put forth are interesting, the claims made by the authors are not fully substantiated by their data and much has been published elsewhere.

1) Authors report that STING colocalized with the CGN after 10 mins, and the TGN after 30 mins by PALM (Figure 1 J, M). This reviewer is having difficulty distinguishing between time points with chosen images. Further, in Figure S1B, STING appears to colocalized with GM130 and TGN38 to the same extent over a time course up to 120 mins.

Thank you for the critical comments.

Our purpose of the experiments (Figure S1B in the original manuscript) was to show that mEos4b-STING was functional in terms of translocation. So we used Zeiss airy-scan microscopy, the spatial (XY) resolution of which is about 120 - 250 nm. With this resolution, we cannot discriminate CGN (GM130) and TGN (TGN38). On the other hand, the spatial (XY) resolution of the PALM image is about 20 - 30 nm. With this resolution, as shown in the revised Figure 2, CGN (GM130) and TGN (TGN38) can be segregated well. With this critical comment, we realized that Figure S1B in the original manuscript was confusing in the context of the present study. We thus deleted this Figure.

We deleted two yellow arrowheads in the original Figure 1J (10 min), which did not show a good co-localization between STING and GM130. The revised one is shown in Figure 2a.

Please be noted that in Figure 1J and M in the original manuscript, that GM130 (30 min) is dim because GM130 is not on the focal plane where STING localizes at this time. The opposite is true for STING (10 min). STING is not on the focal plane where TGN38 localizes at this time.

2) In Figure 2 A/B, the effects of 2-BP and H-151 have largely been shown by previously published works by the authors and others (PMID: 27324217, PMID: 33615891)

We believe that showing the inhibitory effect of these agents on "mEos4b"-STING is still meaningful, therefore we would like to keep them in the manuscript.

3) Figure 2C and H are too dark to evaluate on a screen or paper.

We agree with the reviewer's comment and raised the brightness and contrast of the images. Furthermore, these images were expanded, and shown in Figure 4c and h in the revised manuscript. Please be noted that images should be dim in cells in which STING clustering is suppressed (for example, Figure 6a: D-Cer-C6 vs L-Cer-C6).

4) The authors demonstrated the importance of STING C88/91S in their previous work PMID: 27324217

As related to the critique 2, we believe that showing the result with "mEos4b"-STING is still meaningful, therefore we would like to keep it in Figure 4h in the revised manuscript.

5) For COPA deficient cells, some reports indicate that ER-to-golgi trafficking is also impaired since COPII cannot be recycled back to the ER. Please discuss.

Thank you for the suggestion. We revised the texts accordingly (page 9 line 19).

6) In figure 3A, the author show that 25HC can dampen pTBK1, pSTING and pIRF3. The authors did not include the concentration of 25HC in their methods or how it was delivered to cell culture. Additionally, they did not validate that 25HC was able to influence total or synthesized pools of cholesterol in their model system. For example, how would imaging performed in Fig 3 J be altered with 25-HC?

Thank you for the critical comment. We simply added 30 μ M of 25-HC into the cell culture

medium. We revised the corresponding text in the method section accordingly.

We analyzed the cholesterol content by thin layer chromatography. As shown in the new Figure S10a, 25-HC treatment drastically reduced the total cellular content (compare lane 3 and 4 from the left).

iD4H (the cholesterol-binding probe) may bind 25-HC. Therefore, we performed the experiments with OSW-1. The results in the revised Figure S11d and e showed that the OSW-1 treatment indeed reduced the cholesterol levels in the Rab6-positive area.

7) How do CH25H KO MEFs/macrophage respond to STING ligands?

Thank you for the suggestion. We tried to generate KO MEFs with the CRISPR-Cas9, but failed. We would like to reserve this issue for the future studies.

8) In 3B, authors use OSW-1 to block cholesterol transport from the ER to the golgi, which lowers sting-mediated pTBK/pIRF3. Does this result in build up of cholesterol at the ER? Is OSW toxic to cells or induce apoptosis? What concentrations are utilized?

Thank you for the comments. We used OSW-1 at 0.125 nM, which is now described in the text. With this concentration, we did not see cell death/apoptosis.

Regarding the cholesterol build-up at the ER, we semi-purified the ER membrane (PMID: 26987071) and measured the cholesterol content. As shown, the levels of cholesterol in the ER membrane [cholesterol (μg)/ protein (μg) in the ER membrane] did not differ with /without OSW-1 treatment. Cholesterol may be transported to other organelles (for example, the plasma membrane) from the ER with the aid of other OSBP family members (PMID: 34004151).

9) The use of in vivo OSW is a bit concerning. What cells do the authors think they are

examining from the spleen? I would suspect that these cells are mostly T and B cells with a small portion being innate immune cells. Perhaps a better tactic would be to look at peritoneal macrophage? Or sorting cells from the spleen to exclude adaptive immune cells. For example, OSW may just be killing a population of immune cells from the spleen.

Thank you for the critical and valuable suggestions. We realized that much should be carefully performed to assess the *in vivo* effect of OSW-1. Therefore we removed this data from the manuscript, and would like to reserve this issue for the future studies.

10) Figure 3D is impossible to interpret because too dark

We agree with the reviewer's comment and raised the brightness and contrast of the images. Furthermore, these images were expanded, and shown in Figure 4c in the revised manuscript.

11) The cell free cholesterol system in Figure 3i is misleading and not relevant in this reviewer's opinion. Why not feed cholesterol back to live cells with MβCD? Presumably you would be able to use your Neon Green tracker in figure 3J-M to determine cholesterol uptake into cells.

Thank you for the suggestion. As reviewer suggested, we performed the add-back experiments in live cells. As shown in Figure S9d in the revised manuscript, cholesterol add-back with MβCD beautifully restored the activation of the STING pathway. The cholesterol levels in microsomal fraction were estimated in Figure S10a (lane1 and 2 from the right). We still keep the results of *in vitro* assay (Figure 7g-h in the revised manuscript), because we believe that these two methods complementary support the role of cholesterol in the STING activation.

12) Is the increase in NeonGreen from lipid droplet formation? Or free cholesterol? Can it distinguish between cholesterol and cholesteryl-esters?

Thank you for the interesting suggestion. For the former question, we stained the lipid droplet with LipiBlue dye and found that no significant increase of LipiBlue after STING stimulation

(Figure S12 in the revised manuscript). Therefore, lipid droplet may not be involved in the cholesterol enrichment in the TGN membranes (Figure 7i-k in the revised manuscript). We do not know whether mNeonGreen-iD4H recognizes cholesteryl esters.

13) Authors describe filipin staining in their methods, but I do not find it in their figures or in referenced in their text elsewhere.

Thank you for the comment. This was our mistake. In the revised experiments that other reviewers asked us to do, we used filipin for cholesterol staining (Figure S10b in the revised manuscript).

14) I think that further explanation of some of the methods/reagents and whats already known about STING oligomerization might provide more context for this work.

Thank you for the suggestions. We revised the corresponding texts in Discussion section (page 15).

REVIEWERS' COMMENTS

Reviewer #1 (Remarks to the Author):

The authors addressed all my concerns, no further comments.

Reviewer #2 (Remarks to the Author):

The authors have addressed the reviewers' concerns adequately.

Reviewer #3 (Remarks to the Author):

The authors have mostly addressed my comments and concerns. At this point, this reviewer is satisfied with the authors conclusions.